# Optimal Quantum Speedups for Repeatedly Nested Expectation Estimation

**Yihang Sun** [1]   **Guanyang Wang** [2]   **Jose Blanchet** [1]

## Abstract

We study the estimation of repeatedly nested expectations (RNEs) with a constant horizon (number of nestings) using quantum computing. We propose a quantum algorithm that achieves $\varepsilon$-error with cost $\tilde{O}(\varepsilon^{-1})$, up to logarithmic factors. Standard lower bounds show this scaling is essentially optimal, yielding an almost quadratic speedup over the best classical algorithm. Our results extend prior quantum speedups for single nested expectations to repeated nesting, and therefore cover a broader range of applications, including optimal stopping. This extension requires a new derandomized variant of the classical randomized Multilevel Monte Carlo (rMLMC) algorithm. Careful de-randomization is key to overcoming a variable-time issue that typically increases quantized versions of classical randomized algorithms.

## 1. Introduction

Optimal stopping problems appear ubiquitously in statistics, sequential decision-making, and mathematical finance. Concretely, consider a finite-horizon, discrete-time process $(y_1, \ldots, y_T)$ that users can simulate, the objective is to numerically compute the "optimal utility"

$$V_0 := \sup_{\tau \in \mathcal{T}_T} \mathbb{E}[f(y_\tau)],$$

where $\mathcal{T}_T$ is the set of stopping times (strategies) taking values in $\{1, 2, \ldots, T\}$. By standard optimal stopping theory, we define the Snell envelope as

$$V_k := \sup_{\tau \in \mathcal{T}_{k,T}} \mathbb{E}[f(y_\tau) \mid \mathcal{F}_k], \qquad k = 0, \ldots, T,$$

where $\mathcal{T}_{k,T}$ denotes stopping times $\tau$ satisfying $k \leq \tau \leq T$, and $\mathcal{F}_k$ is the natural filtration. Then $V_k$ satisfies the

dynamic programming recursion

$$V_T = f(y_T), \ V_k = \mathbb{E}[\max\{f(y_k), V_{k+1}\} \mid \mathcal{F}_k] \ k \leq T-1.$$

Optimal stopping is a canonical example of *repeatedly nested expectations* (RNEs) (Syed & Wang, 2023; Rainforth et al., 2018). Beyond optimal stopping, RNEs also arise in applications such as risk estimation for credit valuation adjustment and inference in probabilistic programs; see Syed & Wang (2023) for further examples.

In the classical setting where the number of nestings (the horizon) is fixed, Syed & Wang (2023) proposes a Monte Carlo algorithm with $O(\varepsilon^{-2})$ sample complexity to produce an estimator with mean-squared error $\varepsilon^2$ under smoothness assumptions. The algorithm is developed within the randomized Multilevel Monte Carlo (rMLMC) framework Rhee & Glynn (2015). The $O(\varepsilon^{-2})$ scaling is optimal in its dependence on $\varepsilon$ among all classical Monte Carlo estimators.

This paper studies the estimation of RNEs using quantum computing. Our main contribution is a new quantum algorithm that estimates an RNE with cost $\tilde{O}(\varepsilon^{-1})$, where $\tilde{O}(\cdot)$ ignores logarithmic factors and the dependence on other problem parameters. Therefore, our quantum algorithm achieves an almost quadratic speedup over the optimal classical algorithm. Moreover, standard quantum lower bounds show that this complexity is optimal up to lower-order terms.

Our results extend both the scope and the technical depth of existing work. Blanchet et al. (2026) develops an $\tilde{O}(\varepsilon^{-1})$ method for estimating a single nested expectation. We extend this to multiple repeated nestings using new technical tools described below. This generalization captures a much wider range of applications, including optimal stopping.

From a technical perspective, we develop a new *derandomized* version of the classical rMLMC algorithm used in Syed & Wang (2023), rather than directly quantizing it. A direct quantization strategy would not lead to a quadratic speedup. The key issue is that rMLMC is inherently a *variable-time* algorithm: its random truncation level induces a random runtime across executions. In the quantum setting, such runtime variability interacts poorly with amplitude amplification and can eliminate the intended quadratic speedup; see, e.g., Ambainis (2012). We therefore take a detour: we first derandomize rMLMC by replacing the random runtime

---

[1]Department of Management Science and Engineering, Stanford University, USA [2]Department of Statistics, Rutgers University, USA. Correspondence to: Yihang Sun <kimisun@stanford.edu>.

*Proceedings of the 43$^{rd}$ International Conference on Machine Learning*, Seoul, South Korea. PMLR 306, 2026. Copyright 2026 by the author(s).

with a controlled, deterministic schedule, while maintaining the desired mean-squared error guarantee. We then quantize this derandomized procedure using standard quantum mean-estimation subroutines, yielding $\tilde{O}(\varepsilon^{-1})$ complexity.

## 1.1. Related Works

Our algorithm relies on the quantum mean estimation subroutine of Montanaro (2015); Kothari & O'Donnell (2023), built on Grover's algorithm (Grover, 1996). Related quantum methods also exist. For example, An et al. (2021) proposes a quantum-accelerated multilevel Monte Carlo (MLMC) method, but it does not apply in our setting because our target quantity has a repeatedly nested structure rather than a single telescoping MLMC decomposition. Doriguello et al. (2022) develops a quantum version of the least-squares Monte Carlo (LSM) method (Longstaff & Schwartz, 2001) for optimal stopping. Since LSM approach uses linear regression to approximate the continuation value function, its overall error includes an approximation component that depends on the chosen linear model and can only be bounded under additional function-approximation assumptions. In contrast, our algorithm is essentially non-parametric and targets the general RNE problem, with optimal stopping as a special case.

## 1.2. Setup

Suppose we have an underlying discrete process $y = (y_0, \ldots, y_D) \sim \pi$ for a constant horizon $D \in \mathbb{N}$. Define notation $y_{\le d} = (y_0, \ldots, y_d)$ and $y_{<d} = (y_0, \ldots, y_{d-1})$.

Given functions $g_d : \mathbb{R}^{d+2} \to \mathbb{R}$ for each $0 \le d \le D-1$ and $g_D : \mathbb{R}^{D+1} \to \mathbb{R}$, we want to estimate

$$\gamma_d(y_{<d}) := \mathbb{E}_{y_d}\left[ g_d\left(y_{\le d}, \gamma_{d+1}(y_{\le d})\right) | y_{<d} \right], \quad (1)$$

for $0 \le d < D$ and define $\gamma_D(y_{<D}) = \mathbb{E}_{y_D}[g_D(y)|y_{<D}]$.

**Definition 1.1.** We assume the *last-component bounded Lipschitz* condition (LBL) case, that

1. We can simulate one step of the trajectory at cost 1 and evaluate functions at cost 0.

2. For each $d$, we are given $L_d$ such that $z \mapsto g_d(y_{<d}, z)$ is $L_d$-Lipschitz for $\pi$-almost every $y_{<d}$.

3. $g_D \in L^2(\pi)$, i.e. $\mathbb{E}_\pi |g_D(y)|^2 < \infty$.

This in particular covers the case in Zhou et al. (2023) where $g_d(y_{\le d}, z) = \max\{f(y_{\le d}), z\}$ for a Lipschitz reward function $f$. We henceforth treat $d, D, L_0, \ldots, L_D$ as constants in our asymptotic notation as $\varepsilon \to 0$.

In this setting, Syed & Wang (2023) used a classic randomized Multilevel Monte Carlo (rMLMC) method to obtain the following guarantees on the recursive Algorithm 1.

**Theorem 1.2** (Theorem 2.4 in Syed & Wang (2023))**.** *Under LBL assumptions (Definition 1.1) and any $\delta \in (0, 1/2)$, for each $0 \le d \le D$, there exists $r_d \in (0, 1/2)$ such that given $\pi$-a.e. history $y_{<d}$ and $\mathbb{P}_d = \text{Geo}(r_d)$, Algorithm 1 satisfies*

1. *Unbiased: the expected output is $\gamma_d(y_{<d})$,*

2. *Cost: the expected sample complexity is $O(1)$,*

3. *Moment: the output has $p_d$-th moment at most $O(1)$ where $p_d := 2 - \delta 2^{-d}$.*

---

**Algorithm 1** Algorithm for RNE in Syed & Wang (2023)

**Input:** Time step $d \in \{0, 1, \ldots, D\}$, history $y_{<d} \in \mathbb{R}^d$, and distribution $\mathbb{P}_d$ on $\mathbb{N}$
**Output:** Single sample estimator of $\gamma_d(y_{<d})$

1 Sample $y_d \sim \pi \mid y_{<d}$
2 **if** $d = D$ **then**
3     **return** $g_D(y_{\le D})$
4 **else**
5     Sample $N \sim \mathbb{P}_d$
6     Recursively call the algorithm to estimate $\gamma_{d+1}(y_{\le d})$ for $2^N$ times; label the outputs $X_i$
7     **return** $\Delta_d^{(N)}/\mathbb{P}_d(N)$ where $\Delta_d^{(0)} := g_d(y_{\le d}, X_1)$ and for $n \ge 1$

$$\Delta_d^{(n)} := g_d\left(y_{\le d}, 2^{-n}\sum_{i=1}^{2^n} X_i\right)$$
$$- \frac{1}{2}g_d\left(y_{\le d}, 2^{1-n}\sum_{\text{odd } i=1}^{2^n} X_i\right)$$
$$- \frac{1}{2}g_d\left(y_{\le d}, 2^{1-n}\sum_{\text{even } i=1}^{2^n} X_i\right).$$

---

By calling Algorithm 1 repeatedly on the same history $y_{<d}$ and taking the mean of the estimates, Syed & Wang (2023) bootstraps Theorem 1.2 to obtain the following bound.

**Corollary 1.3** (Corollary 2.5 in Syed & Wang (2023))**.** *For any $\varepsilon > 0$, there is a rMLMC algorithm that estimates $\gamma_d(y_{<d})$ with $L^{p_d}$-error at most $\varepsilon$ and sample complexity $O(\varepsilon^{-2(1+\delta/2^d)})$ as $\varepsilon \to 0$.*

## 1.3. Main Results and Organization

We provide a series results with the same shape as Theorem 1.2 and Corollary 1.3 in both the classical (Section 2) and quantum (Section 3) settings.

First, we will truncate the random level $N$ in Algorithm 1 at the cost of a small bias we control, showing the bounds

in Theorem 1.2 hold for Algorithm 1 with a truncated geometric distribution $\mathbb{P}_d$. The level at which we truncate is $B_d = \Theta(\log(\varepsilon^{-1}))$. This is given as Proposition 2.2.

Next, we replace the random level with a natural deterministic schedule, and obtain the same guarantees. The random truncation version in Proposition 2.2 is a crucial bridge for this step. This answers a question of Syed & Wang (2023) and is our main contribution on the classical-side.

**Theorem 1.4.** *Under LBL assumptions (Definition 1.1) and any $\delta \in (0, 1/2)$, there exists a MLMC algorithm for each $0 \leq d \leq D$ with deterministic level scheduling that, given $\pi$-a.e. history $y_{<d}$ and $\varepsilon > 0$, estimates $\gamma_d(y_{<d})$ with $L^{p_d}$-error at most $\varepsilon$ where $p_d := 2 - \delta/2^d$ and sample complexity $O(\varepsilon^{-2(1+\delta/2^{d-1})})$ as $\varepsilon \to 0$.*

In Section 3, we move to the quantum setting. We first introduce Quantum-Accelerated Monte Carlo (QAMC). Then, we study a natural quantum rMLMC algorithm from directly quantizing the classical algorithm in Syed & Wang (2023), and demonstrate appropriate control on the bias and RMSE.

However, we fall short at the sample complexity control: there is no notion of expected cost for quantum algorithms over a distribution of multiple branches, and we simply pick up the largest cost over all branches. This causes us to pick up a factor of the QAMC cost per time-step remaining.

**Proposition 1.5.** *Under LBL assumptions (Definition 1.1), there is a natural quantization of the classical RNE algorithm for each $0 \leq d \leq D$ that, given $\pi$-a.e. history $y_{<d}$ and $\varepsilon > 0$, estimates $\gamma_d(y_{<d})$ with RMSE at most $\varepsilon$ and sample complexity $\Omega(\varepsilon^{-D+d-1} \log^{D-d+1}(\varepsilon^{-1}))$ as $\varepsilon \to 0$.*

As discussed for example in Ambainis (2012), this is a common issue with direct quantizations of classical variable-time algorithms. However, the derandomized versions of the classical algorithms presented in Section 2 do admit quantization, at the cost of some logarithmic factors per time step. Nonetheless, as $D$ is treated as a constant, the total sample complexity is $\tilde{O}(\varepsilon^{-1})$, giving a quadratic speedup of Corollary 1.3 and Theorem 1.4. This is our main contribution on the quantum-side.

**Theorem 1.6.** *Under LBL assumptions (Definition 1.1), there exists a quantum MLMC algorithm for each $0 \leq d \leq D$ that, given $\pi$-a.e. history $y_{<d}$ and $\varepsilon > 0$, estimates $\gamma_d(y_{<d})$ with RMSE at most $\varepsilon$ and sample complexity at most $O\left(\varepsilon^{-1} \log^{3(D-d)+1}(\varepsilon^{-1})\right)$ as $\varepsilon \to 0$.*

A few remarks are in order:

- Our algorithm uses deterministic level scheduling similar to Theorem 1.4 and Blanchet et al. (2026). The latter corresponds to the $D = 2$ case in Theorem 1.6.

- It obtains quadratic speedup over the classical algo-

rithms in Syed & Wang (2023) via QAMC, and is quantum-inside-quantum in the sense of Blanchet et al. (2026). This is optimal up to log-factors: the RNE problem includes as a special case Monte Carlo mean estimation, which is known to have $\tilde{\Omega}(\varepsilon^{-1})$ quantum complexity (Nayak & Wu, 1999; Hamoudi, 2021).

- Compared to Syed & Wang (2023), our quadratic speedup is further strengthened by controlling the $L^2$-error (i.e. RMSE) instead of $L^{p_d}$-error for $p_d \in (1, 2)$, thereby enabling us to replace the $\varepsilon^{-O(\delta)}$ term in the complexity by an explicit poly-logarithm factor in $\varepsilon^{-1}$.

- There are three logarithm factors per remaining time step, one from the cost per level, one from the number of levels, and the last from applying QAMC.

The last two observations are especially curious as Theorem 1.6 somehow manages to bypass the arbitrarily small $\delta > 0$ deficiency in both error control and cost featured in all the classical theorems. Essentially, the QAMC subroutine gives us sufficiently more slack that simplifies a lot of the intricacies in parameter choices common for classical MLMC algorithms. We discuss this further in Section 3.3 where we also give the algorithm and sketch the proof.

In Section 4, we discuss the complementary simple numerical simulation. The code can be found in this repository.

## 1.4. Quantum Oracle Model and Complexity

In this section, we specify the quantum computation model and the oracle access assumptions. We use the code-access model of Kothari & O'Donnell (2023) for quantum-accelerated Monte Carlo, which matches the setup of Blanchet et al. (2026) for the single-nesting case.

Specifically, we assume standard source-code access to the classical sampling procedure: the algorithm has access to a Python script, circuit description, or pseudocode that generates samples from $p$, rather than only to a black-box sampler or API returning samples. This assumption is standard in simulation settings and holds in the applications considered here. Given such access, the sampling procedure can be converted into a reversible algorithm with the usual reversible-simulation overhead (Bennett, 1973) and then implemented as a quantum circuit. This yields the state-preparation unitary

$$U : |0\rangle \mapsto \sum_x \sqrt{p(x)} |x\rangle |\text{garbage}_x\rangle, \qquad (2)$$

where each $|\text{garbage}_x\rangle$ is an arbitrary unit vector.

Full access to the generator, rather than only to its output samples, is needed because the quantum algorithm must modify the underlying random generation procedure so that it can be run coherently on chosen quantum superpositions.

In Kothari & O'Donnell (2023), this assumption is described as "having the code." In our setting, the randomized objects supplied to QAMC are the recursive routines and the MLMC difference estimators built from them.

We also assume that one trajectory step and one function evaluation each have unit cost. Following Kothari & O'Donnell (2023), the relevant complexity measure is the coherent implementation cost $C(A)$ of one execution of a randomized subroutine $A$. Thus, each QAMC query has cost $\Theta(C(A))$, not $C(A)^2$. In addition, Kothari & O'Donnell (2023) shows that the gate complexity of QAMC scales as $\Theta(C(A))$.

### 1.5. Notations

Let $[M] = \{1, 2, \ldots, M\}$. Let $y_{\leq d} = (y_0, \ldots, y_d)$ and $y_{<d} = (y_0, \ldots, y_{d-1})$, for each $0 \leq d \leq D$. Let $\mathrm{Geo}(r)$ be the geometric random variable with rate $r$. Let $\pi$-a.e. $y$ denote holding for almost every $y \sim \pi$.

For a random variable $X \sim \pi$, let $\|X\|_p := (\mathbb{E}_\pi |X|^p)^{1/p}$ with expectation taken over all the randomness. By $L^p$-error of estimator $R$ of $X$, we mean $\|R - X\|_p$. Let root-mean-squared error (RMSE) be the $L^2$-error $\|R - X\|_2$ and let mean-squared error (MSE) be $\|R - X\|_2^2$.

We adopt standard asymptotic notation in big-$O$ and $\lesssim$ where we treat $d, D, \delta,$ and $L_0, \ldots, L_D$ as constants with $\varepsilon \to 0$. We also have counterparts with $\Theta, \Omega$ as well as $\tilde{O}$ to indicate the omission of logarithmic factors.

## 2. Classical Algorithms

### 2.1. Truncated rMLMC

In this section, we modify Algorithm 1 to show Corollary 1.3 holds if we replace $N \sim \mathrm{Geo}(r_d)$ with distribution $\mathbb{P}_d$ given by $N \sim \mathrm{Geo}(r_d)$ conditioned on $N \leq B_d$ for some cutoff $B_d = O(\log(\varepsilon^{-1}))$. This serves as a review of Syed & Wang (2023) and a model for Theorems 1.4 and 1.6. The main technical steps are two-fold:

- First, the truncation of levels introduces a bias but whose expectation $\beta_d$ can be recursively bounded provided $B_d$ is sufficiently large.

- Second, instead of having a single-sample estimator that we bootstrap via Monte Carlo repetitions in Corollary 1.3, we incorporate the error $\varepsilon$ as a parameter of the algorithm that we satisfy. This aids the induction argument. Consequently, we pre-compute the number $M_d$ of repetitions at time $d$ to guarantee this error.

Both of these subtleties are unnecessary for Syed & Wang (2023), but they are crucial to derandomizing and quantizing

the MLMC algorithm. This is essentially a combination of Algorithm 1 and Corollary 1.3.

We assume number of nestings $D$, functions $g_1, \ldots, g_D$, and $\delta > 0$ are given, as well as distribution $\mathbb{P}_d$ of the random levels and leading constant of $M_d$ are specified in Proposition 2.2. In particular, we first present a numerical lemma that informs the choice of $r_d$, similar to that in Theorem 2.4 of Syed & Wang (2023). Observe that this choice is quite delicate, as is often the case in classic MLMC design to control both the cost and moment.

**Lemma 2.1.** *Fix any $\delta \in (0, 1/2)$. For any $0 \leq d \leq D$, there exists $r_d \in (0, 1)$ such that*

$$(1-r_d)2^{1+\delta/2^{d+1}} = (1-r_d)^{-1+\delta/2^d}2^{-1-\delta/2^{d+1}} < 1. \tag{3}$$

The proof is a simple observation using intermediate value theorem, and is given in Appendix A. With these choices of parameters, we state the following result.

---

**Algorithm 2** Randomized MLMC Algorithm for RNE

---

**Input:** Time step $d \in \{0, 1, \ldots, D\}$ (implicitly), history $y_{<d}$, distribution $\mathbb{P}_d$ on $\mathbb{N}$, and error parameter $\varepsilon > 0$

**Output:** Estimator $R_d(y_{<d}, \varepsilon)$ of $\gamma_d(y_{<d})$ with $L^{p_d}$ error at most $\varepsilon$ for $p_d := 2 - \delta/2^d$

8  $M_d \leftarrow \Theta(\varepsilon^{-2(1+\delta/2^{d-1})})$ from Proposition 2.2

9  **for** $1 \leq m \leq M_d$ **do**

10    Independently sample $y_d^{(m)} \sim \pi \mid y_{<d}$

11    $y_{\leq d}^{(m)} \leftarrow (y_{<d}, y_d^{(m)})$

12  **if** $d = D$ **then**

13    **return** $\frac{1}{M_D} \sum_{m=1}^{M_D} g_D\left(y_{\leq D}^{(m)}\right)$

14  **else**

15    Define $\Delta_d(y_{\leq d}, 0) := g_d\left(y_{\leq d}, R_{d+1}(y_{\leq d}, 1)\right)$ and for $n \geq 1$ define the successive difference

$$\Delta_d(y_{\leq d}, n) := g_d\left(y_{\leq d}, R_{d+1}(y_{\leq d}, 2^{-n/2})\right)$$
$$- g_d\left(y_{\leq d}, R_{d+1}(y_{\leq d}, 2^{-(n-1)/2})\right) \tag{4}$$

16    **for** $1 \leq m \leq M_d$ **do**

17      Independently sample $N^{(m)} \sim \mathbb{P}_d$

18      $A_d^{(m)} \leftarrow \Delta_d\left(y_{\leq d}^{(m)}, N^{(m)}\right) / \mathbb{P}_d(N^{(m)})$

19    **return** $\frac{1}{M_d} \sum_{m=1}^{M_d} A_d^{(m)}$

---

**Proposition 2.2.** *Under LBL assumptions (Definition 1.1) and any $\delta \in (0, 1/2)$, let $p_d := 2 - \delta/2^d$ and let $r_d$ be chosen as in Lemma 2.1. For each $0 \leq d \leq D$, there exists some $B_d, M_d$ such that*

$$B_d := \Theta\left(\log\left(\varepsilon^{-1}\right)\right),$$
$$M_d := \Theta\left(\varepsilon^{-2\left(1+\delta/2^{d-1}\right)}\right), \tag{5}$$

*and that the following holds: for any time step $d$, $\pi$-a.e. history $y_{<d}$, error $\varepsilon \in (0,1)$, and distribution $\mathbb{P}_d$ that is either the geometric distribution given by*

$$\mathbb{P}_d(n) \propto (1 - r_d)^n, \tag{6}$$

*or its truncation at $B_d$ given by*

$$\mathbb{P}_d(n) \propto (1 - r_d)^n \mathbf{1}\{0 \le n \le B_d\}, \tag{7}$$

*then Algorithm 2 outputs an estimator $R_d(y_{<d}, \varepsilon)$ satisfying*

1. *Bias: the mean $\mu_d(y_{<d}) := \mathbb{E}[R_d(y_{<d}, \varepsilon) | y_{<d}]$ is independent of $\varepsilon$ and the bias satisfies*

$$\beta_d := \|\mu_d(y_{<d}) - \gamma_d(y_{<d})\|_{p_d} \le \frac{\varepsilon}{2}\left(1 - \frac{d}{D}\right),$$

2. *Cost: the expected sample complexity of $R_d(y_{<d}, \varepsilon)$ is*

$$C_d(\varepsilon) = O\left(\varepsilon^{-2\left(1 + \delta/2^{d-1}\right)}\right),$$

3. *Moment: the $p_d$-th moment of $R_d(y_{<d}, \varepsilon)$ satisfies*

$$\sigma_d(\varepsilon) := \|R_d(y_{<d}, \varepsilon) - \mu_d(y_{<d})\|_{p_d} \le \varepsilon.$$

*Together, Algorithm 2 estimates $\gamma_d(y_{<d})$ with $L^{p_d}$-error at most $2\varepsilon$ and sample complexity $O(\varepsilon^{-2(1+\delta/2^{d-1})})$.*

Now, Theorem 1.2 and Corollary 1.3 follow immediately as the geometric case, whereas the truncated case is novel and the foundation of our next algorithms. Both proofs are similar. We begin with a corollary of Cauchy-Schwarz.

**Lemma 2.3.** *For any random variables $X_1, \ldots, X_n$*

$$\mathbb{E}\left[(X_1 + \cdots + X_n)^2\right] \le n\sum_{i=1}^n \mathbb{E}[X_i^2]. \tag{8}$$

Under i.i.d. and mean zero assumptions on $X_i$, we reduce the factor of $n$ to 2 on the right, which will be crucial.

**Theorem 2.4** (von Bahr-Esseen). *If $X_i$ are independent with mean zero and $p \in [1, 2]$, then*

$$\mathbb{E}[|X_1 + \cdots + X_n|^p] \le 2\sum_{i=1}^n \mathbb{E}[|X_i|^p].$$

*Hence, if $X_i$ are iid copies of $X$, then*

$$\|X_1 + \cdots + X_n\|_p \le (2n)^{1/p}\|X\|_p.$$

Of course, one may put better constant $c_p$ and it is increasing in $p$ with $c_2 = 2$, so we use 2 for brevity. The proof and further discussions on this inequality can be found in von Bahr & Esseen (1965).

Now, we sketch the key backwards induction strategy used throughout the paper and defer the details to Appendix B.

*Proof Sketch of Proposition 2.2.* We induct backwards on $d$. The base of $d = D$ holds via naive Monte Carlo. given the $d + 1$ case, we show the case of $d$. The expected sample complexity $C_d$ obeys a recursion which solves to the desired bound: it is the sum of the cost for $M_d$ samples of $N$ and the expected cost over $N$ of running $R_{d+1}$ twice with error roughly $2^{-N/2}$. Now, for the bias, we condition on $N \sim \mathbb{P}_d$ to obtain a telescoping series, so that

$$\mathbb{E}[R_d(y_{<d}, \varepsilon) | y_{<d}] = \sum_{n \in \text{supp}(\mathbb{P}_d)} \mathbb{E}[\Delta_d(y_{\le d}, n) | y_{<d}].$$

If $\mathbb{P}_d = \text{Geo}(r_d)$, we induct to show that the algorithm is unbiased since the series is infinite. For truncated $\mathbb{P}_d$, we have an extra term upon telescoping, so it is biased. We bound $\beta_d$ using the Lipschitz condition of $g_d$ and applying inductive hypothesis regarding the moments with $\varepsilon = 2^{-B_d/2}$.

For the moments bound, we first use Theorem 2.4 on the i.i.d. copies $A_d^{(m)}$ indexed by $m \in [M_d]$, to reduce upper bounding $\sigma_d(\varepsilon)^{p_d}$ to upper bounding $\|\Delta_d(y_{\le d}, n)\|_{p_d}$ for each $n$. Then, we use triangle inequality on (4) to insert $g_d$ evaluated at the true mean $\gamma_{d+1}$, and apply the inductive hypothesis on the moments. Skipping some computation, we sum over $n$ to obtain

$$\sigma_d(\varepsilon)^{p_d} \lesssim L_d^{p_d} M_d^{1-p_d} \sum_{n=0}^{\infty} \mathbb{P}_d(n)^{1-p_d}(2^{-n/2})^{p_d}.$$

With careful analysis, we see that this is at most $\varepsilon^{p_d}$ provided $M_d$ is sufficiently large, per (9). $\qquad\square$

### 2.2. Derandomized MLMC

Now, we derandomize the level scheduling Algorithm 2 to obtain Algorithm 3. It has deterministic levels as in classical MLMCs. Essentially, the absorption of the error parameter into Algorithm 2 allows us to specify the level $N^{(m)}$ of each trial individually, and the truncation of $\mathbb{P}_d$ at $B_d$ allows us choose roughly $M_d \mathbb{P}_d(n)$ many trials at level $n$.

Again, we assume that the number of nestings $D$, functions $g_1, \ldots, g_D$, and $\delta > 0$ are given, as well as the distribution $\mathbb{P}_d(n) \propto (1 - r_d)^n \mathbf{1}\{0 \le n \le B_d\}$ from Proposition 2.2 including $r_d$ and $B_d$, the leading constant of $M_d$ from Proposition 2.5, and $\Delta_d(y_{\le d}, n)$ from (4).

**Proposition 2.5.** *Under LBL assumptions (Definition 1.1) and any $\delta \in (0, 1/2)$, let $p_d := 2 - \delta/2^d$ and let $r_d$ be chosen as in Lemma 2.1. For each $0 \le d \le D$, there exists some $B_d, M_d$ such that*

$$\begin{aligned} B_d &:= \Theta\left(\log\left(\varepsilon^{-1}\right)\right), \\ M_d &:= \Theta\left(\varepsilon^{-2\left(1+\delta/2^{d-1}\right)}\right), \end{aligned} \tag{9}$$

*and that the following holds: for any time step $d$, $\pi$-a.e. history $y_{<d}$, error $\varepsilon \in (0,1)$, and the truncated distribution*

$$\mathbb{P}_d(n) \propto (1 - r_d)^n \mathbf{1}\{0 \le n \le B_d\}, \tag{10}$$

*then Algorithm 3 outputs an estimator $R_d(y_{<d}, \varepsilon)$ satisfying*

1. *Bias: the mean $\mu_d(y_{<d}) := \mathbb{E}\left[R_d(y_{<d}, \varepsilon)|y_{<d}\right]$ is independent of $\varepsilon$ and the bias satisfies*

$$\beta_d := \|\mu_d(y_{<d}) - \gamma_d(y_{<d})\|_{p_d} \leq \frac{\varepsilon}{2}\left(1 - \frac{d}{D}\right),$$

2. *Cost: the expected sample complexity of $R_d(y_{<d}, \varepsilon)$ is*

$$C_d(\varepsilon) = O\left(\varepsilon^{-2\left(1+\delta/2^{d-1}\right)}\right),$$

3. *Moment: the $p_d$-th moment of $R_d(y_{<d}, \varepsilon)$ satisfies*

$$\sigma_d(\varepsilon) := \|R_d(y_{<d}, \varepsilon) - \mu_d(y_{<d})\|_{p_d} \leq \varepsilon.$$

*Together, Algorithm 3 estimates $\gamma_d(y_{<d})$ with $L^{p_d}$-error at most $2\varepsilon$ and sample complexity $O(\varepsilon^{-2(1+\delta/2^{d-1})})$.*

---

**Algorithm 3** Classical MLMC Algorithm for RNE

---

**Input:** Time step $d \in \{0, 1, \ldots, D\}$ (implicitly), history $y_{<d}$, and error parameter $\varepsilon > 0$
**Output:** Estimator $R_d(y_{<d}, \varepsilon)$ of $\gamma_d(y_{<d})$ with $L^{p_d}$ error at most $\varepsilon$ for $p_d := 2 - \delta/2^d$

20   $M_d \leftarrow \Theta(\varepsilon^{-2(1+\delta/2^{d-1})})$ from Proposition 2.5
21   $B_d \leftarrow \Theta(\log(\varepsilon^{-1}))$
22   $r_d \leftarrow$ chosen via Lemma 2.1
23   $\mathbb{P}_d(n) \propto (1 - r_d)^n \mathbf{1}\{0 \leq n \leq B_d\}$
24   **if** $d = D$ **then**
25      **for** $1 \leq m \leq M_D$ **do**
26          Independently sample $y_D^{(m)} \sim \pi \mid y_{<D}$
27          $y_{\leq D}^{(m)} \leftarrow (y_{<D}, y_D^{(m)})$
28          **return** $\frac{1}{M_D} \sum_{m=1}^{M_D} g_D\left(y_{\leq D}^{(m)}\right)$
29   **else**
30      **for** $0 \leq n \leq B_d$ **do**
31          $M_d^{(n)} \leftarrow \lfloor M_d \mathbb{P}_d(n) \rfloor$
32          **for** $1 \leq m \leq M_d^{(n)}$ **do**
33              Independently sample $y_d^{(n,m)} \sim \pi \mid y_{<d}$
34              $y_{\leq d}^{(n,m)} \leftarrow (y_{<d}, y_d^{(n,m)})$
35              $A_d^{(n,m)} \leftarrow \Delta_d\left(y_{\leq d}^{(n,m)}, n\right)$ with $\Delta_d$ as in (4)
36      **return** $\sum_{n=0}^{B_d} \frac{1}{M_d^{(n)}} \sum_{m=1}^{M_d^{(n)}} A_d^{(n,m)}$

---

Now, Theorem 1.4 follows immediately. This proposition resembles Proposition 2.2 closely. We explain the reduction here and defer the detailed proof to Appendix C.

*Proof Sketch.* To reduce to Proposition 2.2, we will show that the bias, sample complexity, and moments are bounded

by the corresponding quantities in Algorithm 2 with truncated $\mathbb{P}_d$, up to constant factors in $\delta, d, D$ and $L_d$. Then, applying that proof of Proposition 2.2 suffices.

For the bias, since $y_d^{(m,n)}$ and $A_d^{(n,m)}$ are iid copies, we get

$$\begin{aligned}
\mu_d(y_{\leq d}) &= \mathbb{E}\left[R_d(y_{\leq d}, \varepsilon)|y_{<d}\right] \\
&= \sum_{n=0}^{B_d} \frac{1}{M_d^{(n)}} \sum_{m=1}^{M_d^{(n)}} \mathbb{E}A_d^{(n,m)} \\
&= \sum_{n=0}^{B_d} \Delta_d(y_{\leq d}, n),
\end{aligned} \tag{11}$$

which is the same as Algorithm 2 with truncated $\mathbb{P}_d$. The same is true for the sample complexity bound.

The only caveat is at line 30 where take the floor $\lfloor M_d \mathbb{P}_d(n) \rfloor$ of the counterpart in Algorithm 2. For the reduction to work, we need to show that this effect is only a constant factor, i.e. $M_d \mathbb{P}_d(n) \geq 1$ for every $0 \leq n \leq B_d$. For our choice of $B_d$, this is done carefully in Appendix C by taking the implicit constant factor in $M_d$ in (9) to be $c = (2L_d)^{2+\delta/2^{d-2}}$. $\quad \square$

## 3. Quantum Algorithms

### 3.1. Quantum-Accelerated Monte Carlo

We introduce the key subroutine using quantum amplitude estimation that drives our quantum speedup. We cite the state-of-the-art version before deriving a useful corollary.

**Theorem 3.1** (Quantum-Accelerated Monte Carlo Kothari & O'Donnell (2023))**.** *Given the code of random variable $X$ with standard deviation $\sigma$, for every $\varepsilon, \delta > 0$ there exists a quantum algorithm with sample complexity at most $O\left(\frac{\sigma}{\varepsilon} \log\left(\frac{1}{\delta}\right)\right)$ that outputs estimator $\hat{X}$ such that*

$$\mathbb{P}\left(|\hat{X} - \mathbb{E}X| > \varepsilon\right) \leq \delta.$$

**Corollary 3.2.** *Suppose $\mathbb{E}[X^2] \leq s^2 < \infty$ is bounded with $s$ given as well as the code of $X$, then for every $\varepsilon > 0$ there exists a quantum algorithm that outputs estimator $\hat{X}$ of $\mathbb{E}X$ with RMSE at most $\varepsilon$ and sample complexity at most*

$$O\left(\frac{s}{\varepsilon} \log\left(\frac{s}{\varepsilon}\right)\right).$$

*Proof Sketch.* We apply Theorem 3.1 to algorithmically compute an estimator $\tilde{X}$ such that

$$\mathbb{P}\left(|\tilde{X} - \mathbb{E}X| > \frac{\varepsilon}{2}\right) \lesssim \frac{\varepsilon^2}{s^2}.$$

Then, we output $\hat{X}$ by clipping $\tilde{X}$ into $[-s, s]$ since we know the true mean $\mathbb{E}X$ lies there. Then, we can show that $\hat{X}$ has the desired RMSE, and its sample complexity is identical to that of $\tilde{X}$. $\quad \square$

As shown, upgrading $(\varepsilon, \delta)$-type error to expected error like RMSE is straightforward if we assume bounded raw moments. Indeed, Blanchet et al. (2024) derives analogs of Theorem 3.1 and Corollary 3.2 with $p$-th raw moment and $L^p$-error for $p \in (1, 2)$, and we follow their proof.

With only bounded central moments instead, e.g. the variance, our proof no longer works since we have no bounds $s$ for $\tilde{X}$. However, such a upgrade is still possible: Algorithm 1 of Sidford & Zhang (2023) does so via a clever trick to compare the QAMC estimate with classical estimates, essentially replacing our bound $s$ with one estimated classically. However, this trick is not necessary for us since we will only apply QAMC to estimate level differences in MLMC, which by design have small mean, so the weaker corollary we presented is sufficient.

### 3.2. The Direct Quantization

We first present a natural direct quantization of the algorithm in Syed & Wang (2023). As in Algorithm 2, we use a geometric random variable to sample the level $N_d$ as in randomized MLMC utilized classically by Syed & Wang (2023) and Zhou et al. (2023), and estimate the expectation over $N_d$. We present the pair of recursive algorithms.

---

**Algorithm 4** Main Recursive Estimator for RNE

---

**Input:** Time step $d \in \{0, 1, \ldots, D\}$, history $y_{<d} \in \mathbb{R}^d$, error parameter $\varepsilon > 0$, distribution $\mathbb{P}_d$ on $\mathbb{N}$
**Output:** Estimator $R_d(y_{<d}, \varepsilon)$ of $\gamma_d(y_{<d})$ with RMSE $\varepsilon$

37 **if** $d = D$ **then**

38     Apply Quantum-Accelerated Monte Carlo (Corollary 3.2) to obtain an estimator $R_D(y_{<D}, \varepsilon)$ of

$$\gamma_D(y_{<D}) := \mathbb{E}\left[g_D(y_{\leq D}) | y_{<D}\right]$$

    with RMSE at most $\varepsilon$

39 **else**

40     For $N_d \sim \mathbb{P}_d$ and $A_d$ from Algorithm 5, apply Quantum-Accelerated Monte Carlo (Corollary 3.2) to obtain an estimator $R_d(y_{<d}, \varepsilon)$ of

$$\mu_d(y_{<d}) := \mathbb{E}_{y_d, N_d}\left[\left.\frac{A_d(y_{\leq d}, N_d)}{\mathbb{P}_d(N_d)}\right| y_{<d}\right]$$

    with RMSE at most $\varepsilon$

41 **return** $R_d(y_{<d}, \varepsilon)$

---

As before, we define the bias of $R_d$ and second moment of the object we run QAMC on in $R_d$:

$$\beta_d^2 := \mathbb{E}\left|\mu_d(y_{<d}) - \gamma_d(y_{<d})\right|^2$$

$$\sigma_d(y_{<d})^2 := \mathbb{E}\left[\left.\left|\frac{A_d(y_{\leq d}, N_d)}{\mathbb{P}_d(N_d)}\right|^2\right| y_{<d}\right]$$

---

**Algorithm 5** Auxiliary Estimator for Algorithm 4

---

**Input:** Time step $d \in \{0, 1, \ldots, D - 1\}$, history $y_{\leq d} \in \mathbb{R}^{d+1}$, level $n \in \mathbb{N}$, accuracy rate $\alpha > 0$
**Output:** Estimator $A_d(y_{\leq d}, n)$

42 Call Algorithm 4 to compute estimators of $\gamma_{d+1}(y_{\leq d})$

43     $Z_n \leftarrow R_{d+1}\left(y_{\leq d}, \alpha^n\right)$

44     $Z_{n-1} \leftarrow R_{d+1}\left(y_{\leq d}, \alpha^{n-1}\right)$

45 **return** $g_d(y_{\leq d}, Z_n) - g_d(y_{\leq d}, Z_{n-1})$

---

We analyze the algorithms to obtain the following guarantees, from which Proposition 1.5 follows as the RMSE guarantee holds by the guarantee of the QAMC subroutine.

**Proposition 3.3.** *For $r = 1/2, \alpha = 2/3$, define truncation*

$$B_d := \log_\alpha\left(\frac{\varepsilon}{\sqrt{D}L_0 \ldots L_d}\right). \tag{12}$$

*For any $0 \leq d \leq D$ and $\varepsilon > 0$, with distribution*

$$\mathbb{P}_d(n) \propto (1 - r)^n \mathbf{1}\{0 \leq n \leq B_d\}, \tag{13}$$

*Algorithm 4 on $\pi$-a.e. history $y_{<d}$ returns an estimator $R_d(y_{<d}, \varepsilon)$ satisfying*

1. *Bias: $\beta_d \leq \varepsilon$,*

2. *Moment: $\sigma_d(y_{<d}) = O(1)$,*

3. *Sample complexity is*

$$C_d(\varepsilon) = \Omega(\varepsilon^{D-d+1} \log^{D-d+1}(\varepsilon^{-1})).$$

We remark that due to the variable time issue we alluded to, the per level cost is as if we always pick the most costly level $N_d = B_d$. Then, $R_d(y_{<d}, \varepsilon)$ demands $\tilde{\Theta}(\varepsilon^{-1})$ samples of $R_d(y_{<d}, \alpha^{B_d})$, and $\alpha^{B_d} \asymp \varepsilon$. Hence, the cost recursion solves to $\tilde{\Theta}(\varepsilon^{-1})$ to the power of number of time-steps remaining, as written in the proposition.

This is contrasted against what the expected cost would look like with the QAMC speedup, where each of the $\tilde{\Theta}(\varepsilon^{-1})$ samples has expected cost that of level $N_d$, averaged for $N_d \sim \mathbb{P}_d$. We remark that this cost recursion would solve to

$$\frac{c^{D-d}L_d \ldots L_D}{\varepsilon} \log\left(\frac{L_d}{\varepsilon}\right) \prod_{\ell=d+1}^{D} \log(L_\ell) = \tilde{O}(\varepsilon^{-1})$$

associated with $R_d(y_{<d}, \varepsilon)$, for some constant $c > 0$. This shows that the variable-time issue is the root cause of the failure of the sample complexity bound for Algorithm 4.

### 3.3. Deterministic Level Scheduling

Finally, we give our main quantum algorithm with a fixed schedule for the levels and recover the quadratic speedup

with the worst case cost of $\tilde{\Theta}(\varepsilon^{-1})$. This is a direct generalization of Blanchet et al. (2026) from number of nestings $D = 2$ to arbitrary constant $D$.

---

**Algorithm 6** Quantum MLMC Algorithm for RNE

---

**Input:** Time step $d \in \{0, 1, \ldots, D\}$, history $y_{<d} \in \mathbb{R}^d$, and error parameter $\varepsilon > 0$
**Output:** Estimator $R_d(y_{<d}, \varepsilon)$ of $\gamma_d(y_{<d})$ with RMSE $\varepsilon$
46  $B_d \leftarrow 2 \log_2(L_d/\varepsilon)$
47  **if** $d = D$ **then**
48  $\quad$ Apply Quantum-Accelerated Monte Carlo (Corollary 3.2) to obtain an estimator $R_D(y_{<D}, \varepsilon)$ of

$$\gamma_D(y_{<D}) := \mathbb{E}[g_D(y_{\leq D})|y_{<D}]$$

$\quad$ with RMSE at most $\varepsilon$
49  $\quad$ **return** $R_D(y_{<D}, \varepsilon)$
50  **else**
51  $\quad$ **for** $0 \leq n \leq B_d$ **do**
52  $\quad\quad$ Define successive difference

$$\Delta_d(y_{\leq d}, n) := g_d\left(y_{\leq d}, R_{d+1}(y_{\leq d}, 2^{-n/2})\right)$$
53  $$\quad\quad\quad\quad - g_d\left(y_{\leq d}, R_{d+1}(y_{\leq d}, 2^{-(n-1)/2})\right)$$

54  $\quad\quad$ Apply Quantum-Accelerated Monte Carlo (Corollary 3.2) to obtain an estimator $A_d^{(n)}$ of

$$\mathbb{E}[\Delta_d(y_{\leq d}, n)|y_{<d}]$$

$\quad\quad$ with RMSE at most $\varepsilon/3(B_d + 1)$
55  $\quad$ **return** $\sum_{n=0}^{B_d} A_d^{(n)}$

---

**Proposition 3.4.** *Under LBL assumptions (Definition 1.1), for each $0 \leq d \leq D$, any error $\varepsilon > 0$, and $\pi$-a.e. $y_{<d}$, Algorithm 6 outputs an estimator $R_d(y_{<d}, \varepsilon)$ with MSE*

$$\mathbb{E}\left[(R_d(y_{<d}, \varepsilon) - \gamma_d(y_{<d}))^2 \Big| y_{<d}\right] \leq \varepsilon^2.$$

*Moreover, with leading constant in $d$ and $L$ holding uniformly over $\pi$-a.e. $y_{<d}$, the sample complexity is at most*

$$O\left(\frac{1}{\varepsilon} \log^{1+3(D-d)}\left(\frac{1}{\varepsilon}\right)\right). \tag{14}$$

Now, Theorem 1.6 follows immediately. We give the proof in full detail since this is our main result, and parameter choices are much less delicate than the classical counterparts. We return to a discussion of why after the proof.

*Proof.* We apply backwards induction again. The base case of $d = D$ is clear as Corollary 3.2 gives the estimator with RMSE $\varepsilon$ and cost $O(\varepsilon^{-1} \log(\varepsilon^{-1}))$. Assuming the case of

$d + 1$, we show the case of $d$. For ease of notation, we assume all expectations are taken conditional on $y_{<d}$. For MSE, we see that $\mathbb{E}[\Delta_d(y_{\leq d}, n)|y_{<d}]$ telescopes and

$$\mathbb{E}\left[(R_d(y_{<d}, \varepsilon) - \gamma_d(y_{<d}))^2\right]$$

$$\leq 2\mathbb{E}\left[\left(R_d(y_{<d}, \varepsilon) - \sum_{n=0}^{B_d} \mathbb{E}[\Delta_d(y_{\leq d}, n)]\right)^2\right]$$

$$\quad + 2\mathbb{E}\left[\left(\gamma_d(y_{<d}) - \sum_{n=0}^{B_d} \mathbb{E}[\Delta_d(y_{\leq d}, n)]\right)^2\right]$$

$$\leq 2\mathbb{E}\left[\left(\sum_{n=0}^{B_d} A_d^{(n)} - \mathbb{E}[\Delta_d(y_{\leq d}, n)]\right)^2\right]$$

$$\quad + 2\left(\gamma_d(y_{<d}) - \mathbb{E}\left[g_d\left(y_{\leq d}, R_{d+1}\left(y_{\leq d}, 2^{-B_d/2}\right)\right)\right]\right)^2$$

$$\leq 2(B_d + 1) \sum_{n=0}^{B_d} \mathbb{E}\left[\left(A_d^{(n)} - \mathbb{E}[\Delta_d(y_{\leq d}, n)]\right)^2\right]$$

$$\quad + 2L_d^2\left(\mathbb{E}\left[\gamma_{d+1}(y_{\leq d}) - R_{d+1}(y_{\leq d}, 2^{-B_d/2})\right]\right)^2$$

$$\leq \frac{2\varepsilon^2}{9} + 2L_d^2\mathbb{E}\left[\left(\gamma_{d+1}(y_{\leq d}) - R_{d+1}(y_{\leq d}, 2^{-B_d/2})\right)^2\right]$$

$$\leq \varepsilon^2.$$

In the first step, we apply Lemma 2.3 with $n = 2$; in the second step, we use the telescoping series; in the third step, we apply Lemma 2.3 with $n = B_d + 1$ and rewrite $\gamma_d$ by (1) to apply the Lipschitz property of $g_d$; in the fourth step, we use the QAMC guarantee on $A_d^{(n)}$ and Cauchy-Schwarz; and in the last step we apply the inductive hypothesis and recall our choice of $B_d = 2\log_2(L_d/\varepsilon)$ to obtain

$$\mathbb{E}\left[\left(\gamma_{d+1}(y_{\leq d}) - R_{d+1}(y_{\leq d}, 2^{-B_d/2})\right)^2 \Big| y_{<d}\right]$$

$$= \mathbb{E}_{y_d}\left[\mathbb{E}\left[\left(\gamma_{d+1}(y_{\leq d}) - R_{d+1}(y_{\leq d}, 2^{-B_d/2})\right)^2 \Big| y_{\leq d}\right]\right]$$

$$\leq \mathbb{E}_{y_d}\left[2^{-B_d}\right]$$

$$= \frac{\varepsilon^2}{4L_d^2}.$$

To bound the cost, we first bound the second moment of the random variable we apply QAMC to: for $\pi$-a.e. $y_{<d}$

$$\mathbb{E}\left[\Delta_d(y_{\leq d}, n)^2\right]$$

$$\leq L_d^2\mathbb{E}\left[\left(R_{d+1}(y_{\leq d}, 2^{-n/2}) - R_{d+1}(y_{\leq d}, 2^{-n/2})\right)^2\right]$$

$$\leq 2L_d^2\mathbb{E}\left[\left(R_{d+1}(y_{\leq d}, 2^{-n/2}) - \gamma_{d+1}(y_{\leq d})\right)^2\right.$$

$$\left. + \left(R_{d+1}(y_{\leq d}, 2^{-(n-1)/2}) - \gamma_{d+1}(y_{\leq d})\right)^2\right]$$

$$\leq 2L_d^2(2^{-n} + 2^{1-n})$$

$$= 6L_d^2 \cdot 2^{-n}.$$

which is at most $s^2$ for $s = 3L_d 2^{-n/2}$. In the first step, we apply Lipschitz condition of $g_d$ to (4); the second step

follows Lemma 2.3 with $n = 2$; and the third step follows the inductive hypothesis.

Now, the cost at level $n$ is the sample complexity of QAMC with this second moment bound, times the recursive cost of two calls of $R_{d+1}$ with appropriate accuracy. Therefore, by Corollary 3.2, for $\pi$-a.e. $y_{<d}$, the cost is

$$
\begin{aligned}
C_d(\varepsilon) &\lesssim \sum_{n=0}^{B_d} \frac{s}{\varepsilon/3(B_d+1)} \log\left(\frac{s}{\varepsilon/3(B_d+1)}\right) \\
&\qquad \cdot \left[C_{d+1}(2^{-n/2}) + C_{d+1}(2^{-(n-1)/2})\right] \\
&\lesssim \sum_{n=0}^{B_d} \frac{2^{-n/2} B_d}{\varepsilon} \log\left(\frac{B_d}{\varepsilon}\right) \\
&\qquad \cdot 2^{n/2} \log^{1+3(D-d-1)}\left(2^{n/2}\right) \\
&\lesssim \frac{B_d}{\varepsilon} \log\left(\frac{B_d}{\varepsilon}\right) \sum_{n=0}^{B_d} \left(\frac{n}{2}\right)^{1+3(D-d-1)} \\
&\lesssim \frac{B_d}{\varepsilon} \log\left(\frac{B_d}{\varepsilon}\right) B_d^{3(D-d)-1} \\
&\lesssim \frac{1}{\varepsilon} \log^{3(D-d)+1}\left(\frac{1}{\varepsilon}\right),
\end{aligned}
$$

where the second step follows the inductive hypothesis and the last step follows $B_d \asymp \log(\varepsilon^{-1})$. □

As observed in Section 1.3, our quantum algorithm curiously also bypasses the need for the arbitrarily small deficiency $\delta$ in both the error and the cost: we control the $L^2$-error (i.e. RMSE) instead of $L^{p_d}$-error for $p_d \in (1, 2)$, and replace the $\varepsilon^{-O(\delta)}$ term in the complexity by an explicit poly-logarithm factor in $\varepsilon^{-1}$. We now explain this simplification.

Classically with MLMC or randomized MLMC, we need to tune the geometric ratio in the scheduling of levels to balance both the cost and variance bounds. Here, as is often the case, both are geometric series themselves with reciprocal ratios, so we cannot make both converge simultaneously. This the reason we need the $\delta$-trick in Theorem 1.2 to forego the variance control, and balance delicate calculations in Lemma 2.1 for example.

With Quantum-Accelerated Monte Carlo (Theorem 3.1 and Corollary 3.2), the per-level costs admits a quadratic improvement, so the geometric ratios of cost and variance series are no longer the reciprocals. Then, we have some slack choose some geometric scheduling of the levels to control the convergence of both series.

## 4. Experiments and Simulations

In this repository, we include a simple numerical illustration. It runs two toy-scale experiments: (1) a Quantum-accelerated Monte Carlo building-block test on a credit-risk expected-loss model showing the quantum estimator's query

count scales like $O(\epsilon^{-1})$ versus $O(\epsilon^{-2})$ classically, and (2) an optimal-stopping example verifying the MLMC telescoping differences have variance decaying roughly like $2^{-n}$, which is the structural property our theory relies on.

## Acknowledgements

We thank the reviewers for their discussion and suggestions, especially regarding the inclusion of oracle model discussions as well as numerical simulations.

## Impact Statement

This paper presents work whose goal is to advance the fields of Machine Learning and quantum computing. There are many potential societal consequences of our work and we highlight one specific impact from applications of our algorithm to optimal stopping. More broadly, in mathematical finance, improved computational methods can enhance pricing and risk evaluation. While such improvements may contribute to more accurate and scalable financial modeling, they could also be used to increase the speed and sophistication of trading or automated decision systems. We emphasize that our contribution is methodological and domain-agnostic, and that any broader economic implications depend on how such tools are deployed within existing regulatory and market environments.

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

## A. Proof of Auxiliary Lemmas

*Proof of Lemma 2.1.* By intermediate value theorem applied at $r_d \to 1^-$ and $r_d \to 0^+$, we can find $r_d$ to make the two quantities equal. At equality, it is strictly smaller than one since the product of the two quantities is $(1 - r_d)^{\delta/2^d} < 1$ as $r_d > 0$. $\qquad\square$

*Proof of Corollary 3.2.* Note, standard deviation $\sigma \le s$ and $(\mathbb{E}X)^2 \le \mathbb{E}X^2 \le s^2$, so $|\mathbb{E}X| \le s$. By Theorem 3.1, we can find estimator $\tilde{X}$ such that

$$\mathbb{P}\left(|\tilde{X} - \mathbb{E}X| > \frac{\varepsilon}{2}\right) \le \frac{\varepsilon^2}{8s^2}$$

Now, we clip the estimator into $[-s, s]$, since we know the true mean must lie there, i.e. output

$$\hat{X} = \phi_s(\tilde{X}) \quad \text{where} \quad \phi_s(x) := \min(s, \max(-s, x))$$

Then, $|\hat{X} - \mathbb{E}X| \le 2s$ always, and $\mathbb{P}\left(|\hat{X} - \mathbb{E}X| > \frac{\varepsilon}{2}\right) \le \mathbb{P}\left(|\tilde{X} - \mathbb{E}X| > \frac{\varepsilon}{2}\right) \le \varepsilon^2/8s^2$, so

$$\begin{aligned}
\mathbb{E}(\hat{X} - \mathbb{E}X)^2 &= \mathbb{E}\left[(\hat{X} - \mathbb{E}X)^2 \Big| |\hat{X} - \mathbb{E}X| \le \frac{\varepsilon}{2}\right] \cdot \mathbb{P}\left(|\hat{X} - \mathbb{E}X| \le \frac{\varepsilon}{2}\right) \\
&\quad + \mathbb{E}\left[(\hat{X} - \mathbb{E}X)^2 \Big| |\hat{X} - \mathbb{E}X| > \frac{\varepsilon}{2}\right] \cdot \mathbb{P}\left(|\hat{X} - \mathbb{E}X| > \frac{\varepsilon}{2}\right) \\
&\le \frac{\varepsilon^2}{4} + (2s)^2 \cdot \frac{\varepsilon^2}{8s^2} \\
&\le \varepsilon^2
\end{aligned}$$

so the RMSE is at most $\varepsilon$. Now, the clipping does not cost sample complexity, so by Theorem 3.1, the sample complexity of $\hat{X}$ is that of $\tilde{X}$, which is at most

$$O\left(\frac{\sigma}{\varepsilon/2} \log\left(\frac{1}{\varepsilon^2/8s^2}\right)\right) = O\left(\frac{s}{\varepsilon} \log\left(\frac{s}{\varepsilon}\right)\right)$$

$\qquad\square$

## B. Proof of Proposition 2.2

We induct backwards on $d \in \{0, 1, \ldots, D\}$. Note that $D$ is constant. The base case of $d = D$ is clear: cost $C_D = 1$, the bias $\beta_D = 0$ for both cases of $\mathbb{P}_d$ as $\mu_D(y_{<D}) = \mathbb{E}[g_D(y_{\le D})|y_{<D}] = \gamma_D(y_{<D})$, and $p_D = 2$ so $\sigma_D^2 = \text{Var}[g_D(y_{\le D}) \mid y_{<D}] < \infty$ by Definition 1.1(3).

Now, assuming the $d + 1$ case and we show the $d$ case. First, the expected sample complexity is

$$C_d(\varepsilon) = M_d + M_d \mathbb{E}_{N \sim \mathbb{P}_d}\left[C_{d+1}(2^{-N/2}) + C_{d+1}(2^{-(N-1)/2})\right] \lesssim \varepsilon^{-2(1+\delta/2^{d-1})} \mathbb{E}_{N \sim \mathbb{P}_d}\left[(2^{-N/2})^{-2(1+\delta/2^{d+1})}\right] \quad (15)$$

by absorbing $O(1)$ constants and applying the inductive hypothesis. To show the cost bound, it suffices to bound the expectation in the last term for both $\mathbb{P}_d$. For the geometric case $\mathbb{P}_d(n) = r_d(1 - r_d)^n$ where recall from Lemma 2.1 that we chose $r_d$ to satisfy

$$\rho_d := (1 - r_d)2^{1+\delta/2^{d+1}} = (1 - r_d)^{-1+\delta/2^d} 2^{-1-\delta/2^{d+1}} < 1 \quad (16)$$

Therefore, we compute that

$$\begin{aligned}
\mathbb{E}_{N \sim \mathbb{P}_d}\left[(2^{-N/2})^{-2(1+\delta/2^{d+1})}\right] &= r_d \sum_{n=0}^{\infty} (2^{-n/2})^{-2(1+\delta/2^{d+1})}(1 - r_d)^n \\
&= r_d \sum_{n=0}^{\infty} \left[2^{(1+\delta/2^{d+1})}(1 - r_d)\right]^n \\
&= \frac{r_d}{1 - \rho_d} \\
&\lesssim 1
\end{aligned}$$

as $\rho_d < 1$. For the truncated case, we similarly have

$$
\mathbb{E}_{N \sim \mathbb{P}_d}\left[(2^{-N/2})^{-2(1+\delta/2^{d+1})}\right] = \left(\sum_{n=0}^{B_d}(1-r_d)^n\right)^{-1}\sum_{n=0}^{\infty}(2^{-n/2})^{-2(1+\delta/2^{d+1})}(1-r_d)^n
$$

$$
= \frac{r_d}{1-(1-r_d)^{B_d+1}}\sum_{n=0}^{B_d}\left[2^{(1+\delta/2^{d+1})}(1-r_d)\right]^n
$$

$$
= \frac{r_d}{1-(1-r_d)^{B_d+1}}\cdot\frac{1-\rho_d^{B_d+1}}{1-\rho_d}
$$

$$
\leq \frac{r_d}{1-\rho_d}
$$

$$
\lesssim 1
$$

as $1 - r_d \leq \rho_d < 1$ by Lemma 2.1, so the cost bound holds.

For the bias bound, note that we take $M_d$ iid trials, so for $y_d \sim \pi \mid y_{<d}$, by conditioning on $N \sim \mathbb{P}_d$

$$
\mathbb{E}\left[R_d(y_{<d},\varepsilon)|y_{<d}\right] = \mathbb{E}\left[\frac{\Delta_d(y_{\leq d},N)}{\mathbb{P}_d(N)}\bigg|y_{<d}\right] = \sum_{n\in\mathrm{supp}(\mathbb{P}_d)}\mathbb{E}\left[\frac{\Delta_d(y_{\leq d},N)}{\mathbb{P}_d(N)}\bigg|y_{<d},N=n\right]\mathbb{P}_d(n) = \sum_{n\in\mathrm{supp}(\mathbb{P}_d)}\mathbb{E}\left[\Delta_d(y_{\leq d},n)|y_{<d}\right]
$$

which telescopes by (4). For geometric $\mathbb{P}_d$, we can backwards induct to show Algorithm 2 is in fact unbiased: suppose $\mathbb{E}\left[R_{d+1}(y_{\leq d},\varepsilon')|y_{\leq d}\right] = \gamma_{d+1}(y_{\leq d})$ for any $\varepsilon' > 0$ and $\pi$-almost every $y_{\leq d}$, then for $\pi$-almost every $y_{<d}$

$$
\mathbb{E}\left[R_d(y_{<d},\varepsilon)|y_{<d}\right] = \lim_{n\to\infty}\mathbb{E}\left[g_d\left(y_{\leq d},R_{d+1}(y_{\leq d},2^{-n/2})\right)\bigg|y_{<d}\right]
$$

$$
= \mathbb{E}\left[g_d\left(y_{\leq d},\lim_{n\to\infty}R_{d+1}(y_{\leq d},2^{-n/2})\right)\bigg|y_{<d}\right]
$$

$$
= \mathbb{E}\left[g_d\left(y_{\leq d},\gamma_{d+1}(y_{\leq d})\right)|y_{<d}\right]
$$

$$
= \gamma_d(y_{<d})
$$

as $R_{d+1}(y_{\leq d},2^{-n/2})$ is an unbiased estimator of $\gamma_{d+1}(y_{\leq d})$ with $p_{d+1}$ error $o_{n\to\infty}(1)$ for $\pi \mid y_{<d}$ almost every $y_d$. For truncated $\mathbb{P}_d$, we similarly have

$$
\mathbb{E}\left[R_d(y_{<d},\varepsilon)|y_{<d}\right] = \mathbb{E}\left[g_d\left(y_{\leq d},R_{d+1}(y_{\leq d},2^{-B_d/2})\right)\bigg|y_{<d}\right]
$$

so bias satisfies

$$
\beta_d = \left\|\mathbb{E}\left[g_d\left(y_{\leq d},R_{d+1}(y_{\leq d},2^{-B_d/2})\right) - g_d(y_{\leq d},\gamma_{d+1}(y_{\leq d}))\bigg|y_{<d}\right]\right\|_{p_d}
$$

$$
\leq L_d\left\|R_{d+1}\left(y_{\leq d},2^{-B_d/2}\right) - \gamma_{d+1}(y_{\leq d})\right\|_{p_d}
$$

$$
\leq L_d\left\|R_{d+1}\left(y_{\leq d},2^{-B_d/2}\right) - \gamma_{d+1}(y_{\leq d})\right\|_{p_{d+1}}
$$

$$
\leq 2L_d 2^{-B_d/2}
$$

where we use the induction and that $L^p$ norms are increasing in $p \geq 1$, and $p_d$ is increasing in $d$. Therefore, the bias bound holds by choosing

$$
B_d = 2\log_2\left(\frac{2L_d}{\varepsilon}\right) = \Theta\left(\log\left(\frac{1}{\varepsilon}\right)\right) \tag{17}
$$

as $L_i \geq 1$ for all $i$ by Definition 1.1(2). This proves the bias bound for both cases.

Finally, for the moments, note that conditioned on $y_{<d}$, $A_d^{(m)}$ are iid copies of $\Delta_d(y_{\leq d},N)/\mathbb{P}_d(N)$. Thus, by Theorem 2.4 we have

$$\begin{aligned}
\sigma_d(\varepsilon)^{p_d} &= \|R_d(y_{<d}, \varepsilon) - \mu_d(y_{<d})\|_{p_d}^{p_d} \\
&= \mathbb{E}_\pi \left| \frac{1}{M_d} \sum_{m=1}^{M_d} A_d^{(m)} - \mathbb{E} A_d^{(m)} \right|^{p_d} \\
&\leq 2 M_d^{1-p_d} \mathbb{E} \left| \frac{\Delta_d(y_{\leq d}, N) - \mathbb{E}\Delta_d(y_{\leq d}, N)}{\mathbb{P}_d(N)} \right|^{p_d} \\
&\leq 2 M_d^{1-p_d} \sum_{n=0}^{\infty} \mathbb{P}_d(n)^{1-p_d} \|\Delta_d(y_{\leq d}, n) - \mathbb{E}\Delta_d(y_{\leq d}, n)\|_{p_d}^{p_d} \\
&\leq 2 M_d^{1-p_d} \sum_{n=0}^{\infty} \mathbb{P}_d(n)^{1-p_d} \left( \|\Delta_d(y_{\leq d}, n)\|_{p_d} + \mathbb{E}|\Delta_d(y_{\leq d}, n)| \right)^{p_d} \\
&\leq 8 M_d^{1-p_d} \sum_{n=0}^{\infty} \mathbb{P}_d(n)^{1-p_d} \|\Delta_d(y_{\leq d}, n)\|_{p_d}^{p_d}
\end{aligned}$$

(18)

where in the last step we used $p_d \leq$ and $L^p$ norm is increasing for $p \geq 1$. Now

$$\begin{aligned}
\|\Delta_d(y_{\leq d}, n)\|_{p_d} &\leq \left\| g_d\left(y_{\leq d}, R_{d+1}(y_{\leq d}, 2^{-n/2})\right) - g_d(y_{\leq d}, \gamma_{d+1}(y_{\leq d})) \right\|_{p_d} \\
&\quad + \left\| g_d(y_{\leq d}, \gamma_{d+1}(y_{\leq d})) - g_d\left(y_{\leq d}, R_{d+1}(y_{\leq d}, 2^{-(n-1)/2})\right) \right\|_{p_d} \\
&\leq L_d \left( \|R_{d+1}(y_{\leq d}, 2^{-n/2}) - \gamma_{d+1}(y_{\leq d})\|_{p_d} + \|R_{d+1}(y_{\leq d}, 2^{-(n-1)/2}) - \gamma_{d+1}(y_{\leq d})\|_{p_d} \right) \\
&\leq L_d \left( \|R_{d+1}(y_{\leq d}, 2^{-n/2}) - \gamma_{d+1}(y_{\leq d})\|_{p_{d+1}} + \|R_{d+1}(y_{\leq d}, 2^{-(n-1)/2}) - \gamma_{d+1}(y_{\leq d})\|_{p_{d+1}} \right) \\
&\leq 2 L_d \left( 2^{-n/2} + 2^{-(n-1)/2} \right) \\
&\leq 5 L_d 2^{-n/2}
\end{aligned}$$

Together, we have that

$$\sigma_d(\varepsilon)^{p_d} \leq 200 L_d^{p_d} M_d^{1-p_d} \sum_{n=0}^{\infty} \mathbb{P}_d(n)^{1-p_d} (2^{-n/2})^{p_d}$$

For the geometric case, we have

$$\begin{aligned}
\sigma_d(\varepsilon)^{p_d} &\leq 200 L_d^{p_d} M_d^{1-p_d} \sum_{n=0}^{\infty} r_d^{1-p_d} \left[ (1-r_d)^{1-p_d} 2^{-p_d/2} \right]^n \\
&= 200 L_d^{p_d} (r_d M_d)^{1-p_d} \sum_{n=0}^{\infty} \left[ (1-r_d)^{-1+\delta/2^d} 2^{-1-\delta/2^{d+1}} \right]^n \\
&= 200 L_d^{p_d} (r_d M_d)^{1-p_d} \frac{1}{1-\rho_d}
\end{aligned}$$

and for the truncated case we similarly have

$$\begin{aligned}
\sigma_d(\varepsilon)^{p_d} &\leq 200 L_d^{p_d} M_d^{1-p_d} \left( \sum_{n=0}^{B_d} (1-r_d)^n \right)^{p_d-1} \sum_{n=0}^{B_d} \left[ (1-r_d)^{1-p_d} 2^{-p_d/2} \right]^n \\
&= 200 L_d^{p_d} \left( \frac{r_d M_d}{1-(1-r_d)^{B_d+1}} \right)^{1-p_d} \sum_{n=0}^{B_d} \left[ (1-r_d)^{-1+\delta/2^d} 2^{-1-\delta/2^{d+1}} \right]^n \\
&= 200 L_d^{p_d} \left( \frac{r_d M_d}{1-(1-r_d)^{B_d+1}} \right)^{1-p_d} \left( \frac{1-\rho_d^{B_d+1}}{1-\rho_d} \right) \\
&\leq 200 L_d^{p_d} (r_d M_d)^{1-p_d} \frac{1}{1-\rho_d}
\end{aligned}$$

Therefore, to prove the bound, it suffices to choose $M_d$ such that

$$200 L_d^{p_d} (r_d M_d)^{1-p_d} \frac{1}{1-\rho_d} \le \varepsilon^{p_d}$$

Absorbing $L_d, r_d, \rho_d$ as constant $O(1)$, it suffices to choose $M_d \gtrsim \varepsilon^{-p_d/(p_d-1)}$ which holds by (9) as $(2 - \delta/2^d)/(1 - \delta/2^d) \le 2(1 + \delta/2^{d-1})$. This completes the induction.

Finally, to put it together, the sample complexity is exactly (2) and the $L^{p_d}$-error of $R_d(y_{<d}, \varepsilon)$ is

$$\|R_d(y_{<d}, \varepsilon) - \gamma_d(y_{<d})\|_{p_d} \le \|R_d(y_{<d}, \varepsilon) - \mu_d(y_{<d})\|_{p_d} + \|\mu_d(y_{<d}) - \gamma_d(y_{<d})\|_{p_d} \le \sigma_d(\varepsilon) + \beta_d \le 2\varepsilon$$

## C. Proof of Proposition 2.5

Note that $y_d^{(m,n)}$ are iid copies of $y_d \sim \pi \mid y_{<d}$ for each $m$ and $n$, and $A_d^{(n,m)}$ are iid copies of $\Delta_d(y_{\le d}, n)$ for each $m$, so

$$\mu_d(y_{\le d}) = \mathbb{E} R_d(y_{\le d}, \varepsilon) = \sum_{n=0}^{B_d} \frac{1}{M_d^{(n)}} \sum_{m=1}^{M_d^{(n)}} \mathbb{E} A_d^{(n,m)} = \sum_{n=0}^{B_d} \Delta_d(y_{\le d}, n)$$

which is the same as Algorithm 2 with truncated $\mathbb{P}_d$. This proves the bias bound.

Now, the sample complexity is

$$
\begin{aligned}
C_d(\varepsilon) &= \sum_{n=0}^{B_d} M_d^{(n)} + M_d^{(n)} \left[ C_d(2^{-n/2}) + C_d(2^{-(n-1)/2}) \right] \\
&\le \sum_{n=0}^{B_d} M_d \mathbb{P}_d(n)[1 + C_d(2^{-(n-1)/2} + C_d(2^{-n/2})] \\
&\lesssim \varepsilon^{-2(1+\delta/2^{d-1})} \mathbb{E}_{N \sim \mathbb{P}_d}[1 + C_d(2^{-(n-1)/2}) + C_d(2^{-n/2})]
\end{aligned}
$$

which is the same recursive bound as Algorithm 2, i.e. (15), so the cost bound holds.

Finally, for the moments bound, by Theorem 2.4 again, we have

$$
\begin{aligned}
\sigma_d(\varepsilon)^{p_d} &= \mathbb{E} \left| \sum_{n=0}^{B_d} \sum_{m=1}^{M_d^{(n)}} \frac{A_d^{(n,m)} - \mathbb{E} A_d^{(n,m)}}{M_d^{(n)}} \right|^{p_d} \\
&\le 2 \sum_{n=0}^{B_d} \sum_{m=1}^{M_d^{(n)}} (M_d^{(n)})^{-p_d} \mathbb{E} \left| A_d^{(n,m)} - \mathbb{E} A_d^{(n,m)} \right|^{p_d} \\
&\le 2 \sum_{n=0}^{B_d} (M_d^{(n)})^{1-p_d} \|\Delta_d(y_{\le d}, n) - \mathbb{E}\Delta_d(y_{\le d}, n)\|_{p_d}^{p_d}
\end{aligned}
$$

Compared with (18), to obtain the same bound (up to a constant) as Algorithm 2, it suffices to show for every $n$ that the rounding effect on line 10 of Algorithm 3 is negligible, e.g.

$$(M_d^{(n)})^{1-p_d} = \lfloor \mathbb{P}_d(n) M_d \rfloor^{1-p_d} \le 2 \left( \mathbb{P}_d(n) M_d \right)^{1-p_d}$$

This is true provided $\mathbb{P}_d(n) M_d \ge 1$ for all $n \le B_d$, and so it suffices to show that for our choice of $B_d$ in (17), we can choose constant $c$ so that

$$(1 - r_d)^{B_d} M_d \ge 1 \quad \text{where} \quad M_d = c\varepsilon^{-2(1+\delta/2^{d-1})}$$

Recall $B_d$ from (17), and from Lemma 2.1 we can solve for $1 - r_d$:

$$B_d = 2 \log_2 \left( \frac{2L_d}{\varepsilon} \right) \quad \text{and} \quad 1 - r_d = 2^{-(2+\delta/2^d)/(2-\delta/2^d)}$$

Therefore, we have that

$$
\begin{aligned}
(1 - r_d)^{B_d} M_d &= 2^{-2\left(\log_2\left(\frac{2L_d}{\varepsilon}\right)\right)(2+\delta/2^d)/(2-\delta/2^d)} M_d \\
&= c \left(\frac{2L_d}{\varepsilon}\right)^{-2(2+\delta/2^d)/(2-\delta/2^d)} \varepsilon^{-2(1+\delta/2^{d-1})} \\
&\geq \frac{c}{(2L_d)^{2+\delta/2^{d-2}}} \varepsilon^{2[(2+\delta/2^d)/(2-\delta/2^d)-(1+\delta/2^{d-1})]} \\
&\geq 1
\end{aligned}
$$

by choosing $c = (2L_d)^{2+\delta/2^{d-2}}$ and noting $\varepsilon < 1$ and

$$
\frac{2 + \delta/2^d}{2 - \delta/2^d} < 1 + \frac{\delta}{2^{d-1}}
$$

This proves the moments bound. The conclusion follows the same proof as Proposition 2.2.

## D. Proof of Proposition 3.3

Again, we induct backwards. The base case where $d = D$ is exactly the vanilla quantum-accelerated Monte Carlo Corollary 3.2. Assuming the $d + 1$ case, we show the $d$ case. As the variable-time issue has been extensively discussed in the main body of the paper, we omit the cost bound and show the bias and second moment bounds.

For the $\sigma_d^2$ bound, by the Lipschitz condition of $g_d$

$$
\begin{aligned}
\sigma_d(y_{<d})^2 &= \mathbb{E}\left[\left|\left|\frac{A_d(y_{\leq d}, N_d)}{\mathbb{P}_d(N_d)}\right|\right|^2 \middle| y_{<d}\right] \\
&= \sum_{n=0}^{B_d} \frac{1}{\mathbb{P}_d(n)} \mathbb{E}\left[|A_d(y_{\leq d}, n)|^2 \middle| y_{<d}\right] \\
&= \sum_{n=0}^{B_d} \frac{1}{\mathbb{P}_d(n)} \mathbb{E}\left[\left|g_d(y_{\leq d}, R_{d+1}(y_{\leq d}, \alpha^n)) - g_d(y_{\leq d}, R_{d+1}(y_{\leq d}, \alpha^{n-1}))\right|^2 \middle| y_{<d}\right] \\
&\leq \sum_{n=0}^{B_d} \frac{L_d^2}{\mathbb{P}_d(n)} \mathbb{E}\left[\left|R_{d+1}(y_{\leq d}, \alpha^n) - R_{d+1}(y_{\leq d}, \alpha^{n-1})\right|^2 \middle| y_{<d}\right] \\
&\lesssim \sum_{n=0}^{B_d} \frac{L_d^2}{\mathbb{P}_d(n)} \mathbb{E}\left[\left|R_{d+1}(y_{\leq d}, \alpha^n) - \mu_{d+1}(y_{\leq d})\right|^2 \middle| y_{<d}\right] \\
&\leq \sum_{n=0}^{B_d} \frac{L_d^2 \alpha^{2n}(1 - (1-r)^{B_d})}{r(1-r)^n} \\
&\lesssim \frac{L_d^2(1 - (1-r)^{B_d})(1 - \alpha^{2B_d}(1-r)^{-B_d})}{1 - \alpha^2(1-r)^{-1}} \\
&\leq \frac{L_d^2}{1 - \alpha^2(1-r)^{-1}} \\
&\lesssim L_d^2
\end{aligned}
$$

as $0 < \alpha^2(1-r)^{-1} < 1$ for $r = 1/2, \alpha = 2/3$. Next, we compute the bias. By conditioning on $N_d = n$

$$
\mu_d(y_{<d}) = \sum_{n=0}^{B_d} \mathbb{E}\left[A_d(y_{\leq d}, \alpha^n)|y_d\right] = \mathbb{E}\left[g_d(y_{\leq d}, R_d(y_{\leq d}, \alpha^{B_d}))\big|y_d\right]
$$

Therefore, by the Lipschitz property, we have

$$
\begin{aligned}
\beta_d^2 &= \mathbb{E}|\mu_d(y_{<d}) - \gamma_d(y_{<d})|^2 \\
&= \mathbb{E}_{y_{<d}} \left| \mathbb{E}\left[ g_d(y_{\leq d}, R_{d+1}(y_{\leq d}, \alpha^{B_d})) - g_d(y_{<d}, \gamma_{d+1}(y_{\leq d})) \middle| y_{<d} \right] \right|^2 \\
&\leq L_d^2 \mathbb{E}_{y_{<d}} \mathbb{E}\left[ \left| R_{d+1}(y_{\leq d}, \alpha^{B_d}) - \gamma_{d+1}(y_{\leq d}) \right|^2 \middle| y_{<d} \right] \\
&\leq L_d^2 \mathbb{E}\left[ |\mu_{d+1}(y_{\leq d}) - \gamma_{d+1}(y_{\leq d})|^2 + |R_{d+1}(y_{\leq d}, \alpha^{B_d}) - \mu_{d+1}(y_{\leq d})|^2 \right] \\
&\leq L_d^2 (\alpha^{2B_d} + \beta_{d+1}^2)
\end{aligned}
$$

where the last step follows the inductive hypothesis and definition of the QAMC subroutine. Solving the recursion and recalling our choice of $B_d$, we obtain the following bound

$$
\beta_d^2 \leq \sum_{\ell \geq d}^{D} L_d^2 \ldots L_\ell^2 \alpha^{2B_\ell} \leq \varepsilon^2.
$$

