# OpenReview forum: "Optimal Quantum Speedups for Repeatedly Nested Expectation Estimation"
_ICML.cc/2026/Conference — ICML 2026 regular_

### Official Review · Reviewer_Wzu9 · 2026-03-12

**Soundness:** 3
**Presentation:** 2
**Significance:** 3
**Originality:** 3
**Overall Recommendation:** 4
**Confidence:** 2

**Summary:**

This paper proposes  a quantum algorithm for estimating the repeatedly nested expectations with a computational cost of $\tilde{O}(\epsilon^{-1})$, which is near optimal and achieves quadratic speedup over the classical algorithms.

**Compliance With Llm Reviewing Policy:**

Affirmed.

**Final Justification:**

The authors have addressed my concern. I raise my score to 4.

**Key Questions For Authors:**

1. Although the near optimal complexity is a strong contribution to the repeated nestings estimation, the optimal quantum complexity has already been developed for estimating a single nested expectation. Can the authors state the main difficulty in generalizing the single nested expectation estimation to the repeated nestings expectation estimation.

2. It seems that the contribution is mainly on developing a derandomized MLMC framework.. The optimal quantum MLMC algorithm seems a direct application of quantum accelerated Monte Carlo on the derandomized MLMC framework for RNE. Can the authors state their novelty in quantum computing?

3. The writing and organization of this paper needs to improve.
* Can the author clearly define the $\epsilon$-$L^{p}$-error?
* Can the author clearly state the classical sample oracle and quantum sample oracle? Since the theorems reported the sample complexity, a clear definition on the sample oracle is important.

**Limitations:**

Please refer to the question part.

**Strengths And Weaknesses:**

**Strengths**
* This paper proposes a novel quantum methods for repeated nested expectation estimation. To achieve this, the authors propose derandomized MLMC method and uses quantum Monte Carlo.
* The obtained quantum complexity is near optimal. The theoretical contribution of this paper is strong.

**Weakness**
Please refer to the question part.

---

> ### Author Rebuttal · Authors · 2026-03-31
>
> We thank the reviewer for the valuable suggestions. We also appreciate the recognition of the novelty of our work and its strong theoretical contributions. Below, we summarize and address the questions raised.
>
>
> **The main challenge beyond single nesting**: The main challenges come from the recursive structure of repeated nesting. This makes repeated nesting setup significantly more difficult than single nesting, both classically and in the quantum setting. More precisely:
>
> * For classical algorithm: The key idea of MLMC is to design an estimator that simultaneously controls the computational cost, bias, and variance at the desired levels. In the multiple nesting case, controlling these quantities jointly becomes challenging, since they all propagate across the nesting levels. Therefore, although the single-nesting case is relatively well understood, progress on the multiple-nesting case has only emerged very recently. See, for example, [1] for the triply nested expectation case (described as “extremely challenging” on p.13 of [2]), and [3,4] for repeated nested expectations.
>
> * For quantum algorithm: Quantizing repeated nested expectations is substantially more difficult than quantizing a single nested expectation. In the single-nesting case, classical methods include both randomized and non-randomized MLMC, and thus [5] can quantize the non-randomized version directly. In contrast, for the repeated nesting problem, all existing classical methods rely on variable-time randomized MLMC, which forces us to first develop a derandomized version (for the reasons discussed in our response to the next question). As a result, quantizing repeated nesting requires an additional derandomization step, which in turn demands careful joint control of the bias, variance, and computational cost arising from the recursive structure.
>
>
> We thank the reviewer for this question and will include a more detailed discussion of these challenges in the revised paper.
>
> **Novelty in quantum computing**: We agree that our algorithm uses standard QAMC as a subroutine. The quantum contribution of this paper is not a new mean-estimation algorithm, but rather a new quantum-compatible framework for handling an important class of variable-time algorithms. More precisely,
>
> * Variable-time issue: The variable-time issue is a well-known challenge in quantum computation (see [5] and the references therein). For a classical variable-time algorithm such as randomized MLMC, different computational branches may terminate at different times. However, in the quantum setting, one cannot simply average the computational costs over these random branches which makes the direct quantization of such variable-time algorithms substantially more difficult. Our paper provides a concrete way to overcome this obstacle for RNEs by derandomizing randomized MLMC, thereby achieving a near-optimal guarantee.
>
> * Quantizing other randomized MLMC algorithms: Randomized MLMC is a widely used classical method in both statistics and operations research. Therefore, any attempt to quantize this type of estimator will face the same obstacle. For this reason, we believe the ideas developed here may also be useful beyond RNEs, for other problems that involve quantizing randomized MLMC algorithms.
>
> We will also highlight these technical novelties more clearly in the revised version of the paper. Thank you for raising this question.
>
>
> **Presentation and oracle definition:** Thanks for the question! The $L^p$-error is defined in the notations section, it means $\Vert \hat{X}-X\Vert_p = \left(\mathbb{E}\left[\Vert \hat{X}-X\Vert ^p\right]\right)^{1/p}\le\varepsilon$ with expectation taken over all the randomness. The oracle models are described in the rebuttal for Reviewer mPiR, with corresponding discussions about classical and quantum sampling complexities.
>
> We will also add the oracle definition explicitly in a separate section of the paper.
>
> Ref:
> [1] Efficient Risk Estimation for the Credit Valuation Adjustment, Giles et. al., Finance and Stochastics
>
> [2] MLMC techniques for discontinuous functions, Giles, arXiv:2301.02882
>
> [3] Unbiased optimal stopping via the MUSE, Zhou et. al., Stochastic Processes and their Applications
>
> [4] Optimal randomized multilevel Monte Carlo for repeatedly nested expectations, Syed and Wang, ICML'23
>
> [5] Variable time amplitude amplification and quantum algorithms for linear algebra problems, Ambainis, STACS'12

---

> > ### Author Rebuttal · Reviewer_Wzu9 · 2026-04-01
> >
> > Thanks for your response to my concern. This paper seems technical solid and I decide to raise my score to 4.

---

### Official Review · Reviewer_gHLm · 2026-03-12

**Soundness:** 3
**Presentation:** 2
**Significance:** 3
**Originality:** 3
**Overall Recommendation:** 4
**Confidence:** 2

**Summary:**

This paper studies the estimation of repeatedly nested expectations (RNEs) under a constant nesting horizon $D$ using quantum algorithms. The authors propose an algorithm that achieves sample complexity $\tilde{O}(\varepsilon^{-1})$, which is quadratically better than the classical method. The key insight is that a direct quantization of the classical rMLMC (randomized Multilevel Monte Carlo) estimator does not yield the desired speedup. The paper first truncates and derandomizes the classical schedule and then quantizes the resulting deterministic-level algorithm. This extends prior quantum results for single nested expectations to repeatedly nested ones, and the paper also gives a direct-quantization negative result that helps justify why the final construction is necessary.

**Compliance With Llm Reviewing Policy:**

Affirmed.

**Final Justification:**

After reading the other reviews, I believe this paper makes meaningful contributions, so I have decided to maintain my positive score.

**Key Questions For Authors:**

1. What exact quantum access model is assumed for simulating one step of the process and evaluating the relevant functions? If these primitives are expensive, how should one interpret the stated complexity?
2. Could the authors include a small illustrative example, such as a toy optimal stopping instance, to help readers understand the algorithm and its practical meaning?

**Limitations:**

1. The result relies on a constant nesting horizon $D$.
2. The paper focuses only on sample complexity rather than quantum resource costs such as gate complexity and circuit depth.

**Strengths And Weaknesses:**

Strengths:
1. The technical insight is nontrivial. The authors identify why the direct quantumization fails and redesign the classical estimator to make quantization work.
2. The paper extends prior $\tilde{O}(\varepsilon^{-1})$ quantum results for single nested expectations to repeated nesting with constant horizon, which is a substantially broader class including optimal stopping.
3. The negative result on direct quantization is conceptually valuable, as it clarifies an important obstruction and strengthens the motivation for the proposed algorithm.

Weaknesses:
1. The final bound (Theorem 1.6) has a polylog exponent growing with $D-d$. When the horizon $D$ is large, the performance of the proposed algorithm may degrade quickly.
2.  The paper does not discuss the oracle model in sufficient detail. In particular, it remains unclear what quantum access is required for trajectory simulation and function evaluation, and how the claimed complexity should be interpreted in terms of gate complexity and other quantum resources.
3. The paper is entirely theoretical and quite dense in presentation. Even a very simple numerical illustration or toy optimal stopping example would help make the construction more concrete and easier to appreciate.

---

> ### Author Rebuttal · Authors · 2026-03-31
>
> We thank the reviewer for the valuable suggestions. We also appreciate the recognition of theoretical important to generalize to a constant number of nestings. Below, we summarize and address the questions raised.
>
> **Dependence on the number of nestings:**
> we acknowledge linear dependence on $D$ in the exponent of $\log(1/\epsilon)$ is a limitation of our methods. This is the standard treatment in classical literature as well [1] and moreover several applications in practice have the number $D$ of nestings be a moderate constant, e.g. Bermuda options where $D=3$.
>
> The difficulty to improving this dependence boils down to the variable-time issue described in the paper. It is a well-known challenge in quantum computation (see [5] and the references therein). For a classical variable-time algorithm such as randomized MLMC, different computational branches may terminate at different times, and the direct quantization in Section 3.2 would remove the the linear dependence on $D$ in the exponent of $\log(1/\epsilon)$ if the branch costs are averaged. However, in the quantum setting, one cannot simply average the computational costs over these random branches which makes the direct quantization of such variable-time algorithms substantially more difficult. The derandomization step of the classical algorithm to overcome this obstacle gives rise to the linear dependence on $D$ in the exponent of $\log(1/\epsilon)$. We agree that this issue is a limitation, and overcoming it would be an interesting future direction.
>
> **Experiments and simulations:**
> We add a simple numerical illustration. It runs two toy-scale experiments: (1) a Quantum-accelerated Monte Carlo
> building‑block test on a credit‑risk expected‑loss model showing the quantum estimator’s query
> count scales like $O(\epsilon^{-1})$ (vs classical $O(\epsilon^{-2})$), and (2) an optimal‑stopping example verifying the MLMC telescoping differences have variance decaying roughly like $2^{-n}$, which is the
> structural property our theory relies on. We thank the reviewer for this suggestion and agree a simple toy simulation is possible and aids understanding.
>
> **Quantum oracle model and complexity:**
> We thank the reviewer for this comment and agree that we need to add oracle model explicitly as a separate section of our paper. We assume the have the code model of [2] for the quantum-accelerated Monte Carlo implementation which is the same as the setup in [3] for the quantum algorithm in the case of a single nesting.
>
> Concretely, we assume the standard source-code access model: one has full algorithmic access to the classical sampling procedure.
> By that, we mean access to a Python script or pseudocode that generates samples from $p$, rather than access only through a black-box interface or API that returns samples. This is a standard assumption in the simulation literature, and it holds in all applications we are aware of.
>
> Given the above access, one can convert it into a reversible algorithm with only constant-factor overhead in time complexity [4], and then implement it as a quantum circuit
> $$
> U: |0\rangle \mapsto \sum_x \sqrt{p(x)} |x\rangle |\mathrm{garbage}_x\rangle,
> $$
> where the states $|\mathrm{garbage}_x\rangle$ may be arbitrary unit vectors. This state preparation model is also standard in the quantum computing literature, such as [2, 3].
>
> We require full access to the generator (rather than only to the samples it produces) because the quantum algorithm needs to modify the underlying random generation procedure so that it can be run on carefully chosen quantum superpositions. In [2], this is formalized as ``having the code'': access to a unitary quantum circuit with a fixed input such that, upon measuring its output and discarding some bits, we get a draw from the distribution, which is exact what the unitary $U$ gives in the conversion above. In our setting, the recursive routines and MLMC difference estimators built from them are exactly the randomized objects supplied to QAMC.
>
> We further assume that simulating one step of the trajectory and one function evaluation each have unit cost in our model. As explained in [2], in this model the complexity measure is the coherent implementation cost $C(A)$ of one execution of a randomized subroutine $A$. If one coherent execution of a subroutine $A$ costs $C(A)$, then a call to QAMC still has per-query cost $\Theta(C(A))$.
>
> Ref:
> [1] Optimal randomized multilevel Monte Carlo for repeatedly nested expectations, Syed and Wang, ICML'23
>
> [2] Mean estimation when you have the source code; or, quantum Monte Carlo methods, Kothari and O'Donnell, SODA'23
>
> [3] Non-linear Quantum Monte Carlo, Blanchet, Hamoudi, Szegedy, and Wang, NeurIPS'25
>
> [4] Logical reversibility of computation, Bennett, IBM journal of Research and Development, 1973
>
> [5] Variable time amplitude amplification and quantum algorithms for linear algebra problems, Ambainis, STACS'12

---

> > ### Author Rebuttal · Reviewer_gHLm · 2026-04-02
> >
> > Thanks for the detailed clarification. I do not have any further questions at this time.

---

### Official Review · Reviewer_eFDs · 2026-03-13

**Soundness:** 3
**Presentation:** 3
**Significance:** 3
**Originality:** 2
**Overall Recommendation:** 4
**Confidence:** 4

**Summary:**

The manuscript introduces a quantum algorithm for estimating repeatedly nested expectations (RNEs) with a constant horizon, achieving an $\epsilon$-error with a sample complexity of $\tilde{O}(\epsilon^{-1})$. This provides a near-quadratic speedup over the optimal classical algorithms. To achieve this without the penalties of variable-time quantum execution, the authors first derandomize the classical randomized Multilevel Monte Carlo (rMLMC) algorithm by replacing its random truncation level with a controlled, deterministic schedule. They then quantize this derandomized version using Quantum-Accelerated Monte Carlo (QAMC) subroutines. The work extends previous quantum speedups from single to multiple repeated nestings, making it applicable to broader problems like optimal stopping.

**Compliance With Llm Reviewing Policy:**

Affirmed.

**Key Questions For Authors:**

Here are some questions for authors:

1. How does the algorithm perform if the horizon $D$ is not a strict constant but rather scales slowly with $\epsilon$? Can the dependence on $D$ in the exponent of the logarithmic factors be optimized further to accommodate deeper nestings?

2. The quantization relies heavily on Quantum-Accelerated Monte Carlo (QAMC). Could you discuss the practical hardware assumptions (e.g., fault-tolerance thresholds, qubit depth requirements) necessary to realistically implement these specific QAMC subroutines for optimal stopping problems?

3. A direct quantization strategy picks up a factor of the QAMC cost per time-step remaining. Are there any specific problem structures where the direct quantization of the variable-time algorithm might actually be preferable in the near term, or is the derandomized schedule universally superior?

**Limitations:**

No, the authors have not adequately discussed the limitations and potential negative societal impact of their work.

Constructive Suggestions for Improvement:

1. The authors use a boilerplate impact statement claiming there are "none which we feel must be specifically highlighted here". They should replace this with a substantive discussion. Since optimal stopping problems are heavily utilized in mathematical finance, the authors could discuss the dual-use nature of accelerated financial modeling and its broader economic implications.
2. The authors should dedicate a specific section to synthesizing the technical limitations of their approach. They must explicitly discuss the strict limitation of treating the nesting horizon $D$ as a constant, as well as the substantial gap between the theoretical fault-tolerant assumptions of the QAMC subroutines and the reality of current Noisy Intermediate-Scale Quantum (NISQ) devices.

**Strengths And Weaknesses:**

Strengths:
1. Successfully bridging the gap between single-nested and repeatedly nested expectations in the quantum setting is a strong theoretical contribution. The paper rigorously demonstrates that the scaling is essentially optimal, bounded by standard lower bounds.
2. The identification of the variable-time issue in quantizing standard rMLMC and the proposed solution—derandomizing the classical algorithm first before applying QAMC—is a clever and well-executed algorithmic design choice.
3. The paper provides extensive and solid mathematical proofs for both the classical derandomized bounds (Theorem 1.4) and the quantum bounds (Theorem 1.6).

Weaknesses:
1. The horizon $D$ (number of nestings) is treated strictly as a constant throughout the analysis. The sample complexity scales with logarithmic factors to the power of $3(D-d)+1$, which implies the algorithm's performance would degrade significantly if the horizon $D$ grows.
2. The paper is purely theoretical. While not strictly required for a theory-heavy submission, providing a toy simulation or classically simulating the quantum scaling behavior would help ground the theoretical scaling limits in a practical context.

---

> ### Author Rebuttal · Authors · 2026-03-31
>
> We thank the reviewer for the valuable suggestions. We also appreciate the recognition of the results and strong theoretical contributions. Below, we summarize and address the questions raised.
>
> **Dependence on the number of nestings:**
> we acknowledge linear dependence on $D$ in the exponent of $\log(1/\epsilon)$ is a limitation of our methods. This is the standard treatment in classical literature as well [1] and moreover several applications in practice have the number $D$ of nestings be a moderate constant, e.g. Bermuda options where $D=3$.
>
> The difficulty to improving this dependence boils down to the variable-time issue described in the paper. It is a well-known challenge in quantum computation (see [2] and the references therein). For a classical variable-time algorithm such as randomized MLMC, different computational branches may terminate at different times, and the direct quantization in Section 3.2 would remove the the linear dependence on $D$ in the exponent of $\log(1/\epsilon)$ if the branch costs are averaged. However, in the quantum setting, one cannot simply average the computational costs over these random branches which makes the direct quantization of such variable-time algorithms substantially more difficult. The derandomization step of the classical algorithm to overcome this obstacle gives rise to the linear dependence on $D$ in the exponent of $\log(1/\epsilon)$. We agree that this issue is a limitation, and overcoming it would be an interesting future direction.
>
> **Algorithmic and hardware consideration:**
> We thank the reviewer for pointing out a lack of clarity regarding the quantum computation model and oracle access. As observed by the reviewer, our model boils down to the model in [3] of the quantum-accelerated Monte Carlo implementation. This is the same model taken in [4] for the quantum algorithm in the case of a single nesting.
>
> Concretely, we assume the standard source-code access model: one has full algorithmic access to the classical sampling procedure.
> By that, we mean access to a Python script or pseudocode that generates samples from $p$, rather than access only through a black-box interface or API that returns samples. This is a standard assumption in the simulation literature, and it holds in all applications we are aware of. Given the above access, one can convert it into a reversible algorithm with only constant-factor overhead in time complexity [5], and then implement it as a quantum circuit. We further assume that simulating one step of the trajectory and one function evaluation each have unit cost in our model.
>
> We discuss the model in further detail in the rebuttal to reviewer mPiR. Here, we highlight that our model (that of [3]) assumes a mature fault-tolerant architecture with below-threshold physical operations, fast syndrome decoding, and many logical qubits. While recent progress in hardware has been made, this algorithm should not be viewed as a target for current Noisy Intermediate-Scale Quantum (NISQ) devices. We acknowledge this as a practical limitation and will dedicate a section of the paper to further discuss it.
>
> **Experiments and simulations:**
> We add a simple numerical illustration. It runs two toy-scale experiments: (1) a Quantum-accelerated Monte Carlo
> building‑block test on a credit‑risk expected‑loss model showing the quantum estimator’s query
> count scales like $O(\epsilon^{-1})$ (vs classical $O(\epsilon^{-2})$), and (2) an optimal‑stopping example verifying the MLMC telescoping differences have variance decaying roughly like $2^{-n}$, which is the
> structural property our theory relies on. We thank the reviewer for this suggestion and agree a simple toy simulation is possible and aids understanding.
>
> **The impact statement:**
> We agree with the reviewer on the impact from applications of our algorithm to optimal stopping in mathematical finance, where improved computational methods can enhance pricing and risk evaluation. While such improvements may contribute to more accurate and scalable financial modeling, they could also be used to increase the speed and sophistication of trading or automated decision systems. We emphasize that our contribution is methodological and domain-agnostic, and that any broader economic implications depend on how such tools are deployed within existing regulatory and market environments. We will include this discussion in our impact statement.
>
> Ref:
> [1] Optimal randomized multilevel Monte Carlo for repeatedly nested expectations, Syed and Wang, ICML'23
>
> [2] Variable time amplitude amplification and quantum algorithms for linear algebra problems, Ambainis, STACS'12
>
> [3] Mean estimation when you have the source code; or, quantum Monte Carlo methods, Kothari and O'Donnell, SODA'23
>
> [4] Non-linear Quantum Monte Carlo, Blanchet, Hamoudi, Szegedy, and Wang, NeurIPS'25
>
> [5] Logical reversibility of computation, Bennett, IBM journal of Research and Development, 1973

---

> > ### Author Rebuttal · Reviewer_eFDs · 2026-04-04
> >
> > Thanks to the authors. I will keep my score.

---

### Official Review · Reviewer_mPiR · 2026-03-13

**Soundness:** 3
**Presentation:** 2
**Significance:** 3
**Originality:** 2
**Overall Recommendation:** 4
**Confidence:** 3

**Summary:**

The paper studies the problem of repeatedly nested expectations, of which optimal stopping time is an important example, and proposes a
derandomized variant of the classical randomised Multilevel Monte Carlo algorithm, which is then quantised in order to obtain a
quadratic speed-up in terms of error $\varepsilon$ compared to its classical counterparts. Their derandomised classical algorithm is
obtained by introducing a truncation trick and trading the expectation of some random variables up to a randomised error by the sum of the expectations of these random variables for all possible values of the error (up to a truncation value). Their derandomised algorithm is
shown to be easily quantised by employing the standard quantum Monte Carlo subroutine. In contrast, the authors show that without
introducing such derandomisation, a naive quantisation of the standard classical algorithm would not lead to a full quadratic speed-up.

**Compliance With Llm Reviewing Policy:**

Affirmed.

**Key Questions For Authors:**

In terms of readability, the paper is well written but there are too
many typos (see below). I urge the authors to carefully reread their
paper.

List of corrections:
1. Almost all equations don't have proper punctuation. Please rectify this.
2. Page 2: in "following guarantees on the recursive Algorithm 2" it
should be Algorithm 1.
3. Page 2: "We provide a series results" should be "We provide a
series of results".
4. Page 2: "is crucial bridge" should be "is a crucial bridge".
5. Page 3, Proposition 1.5: I reckon the complexity is
$\Omega(\varepsilon^{-(D-d+1)}\log^{D-d+1}(1/\varepsilon))$ and not
$\Omega(\varepsilon^{D-d+1}\log^{D-d+1}(1/\varepsilon))$.
6. Page 4: explain a bit more what $M_d$ is when first mentioning it.
7. Page 4, Proposition 2.2: $p_d = 2 - \delta/2^d$ and not $p_d = 1 -
\delta/2^d$.
8. Page 4, Proposition 2.2: define what $\mu_d(y_{<d})$ is.
9. Page 4, Theorem 2.4: for $p=2$, the constant $2$ on the right-hand
side of the inequality is not needed, if you haven't noticed.
10. Page 5: the sentence "we use triangle inequality on (3) to insert
$g_d$ evaluated not that recursive output of $R_{d+1}$ but true mean"
reads badly.
11. Page 5, Proposition 2.5: $p_d = 2 - \delta/2^d$ and not $p_d = 1 -
\delta/2^d$.
12. Page 5, Proposition 2.5: "Algorithm 2" should be "Algorithm 3".
13. Page 6: which quantum oracles are you using? What does it mean to
you to have a code of a random variable?
14. Page 6: " in Algorithm 1 of Sidford & Zhang (2023) does so" should
be "Algorithm 1 of Sidford & Zhang (2023) does so".
15. Page 6: "this trick not necessary" should be "this trick is not necessary"
16. Page 8: when you use the telescoping series in the second step,
what happens to the term $n=0$?
17. Page 8: "balance the both cost and variance bounds" should be
"balance both the cost and variance bounds".
18. Page 10: $=\lesssim" in the last line should be "\lesssim 1".
19. Page 14: Corollary 3.2 should probably be Proposition 3.4.
20. Page 14: the fourth equality in the $\sigma_d^2$ bound in the
proof of Proposition 3.3 should be an inequality.
21. Page 15: the fourth equality in the $\beta_d^2$ bound in the proof
of Proposition 3.3 should be an inequality.

**Limitations:**

yes

**Strengths And Weaknesses:**

The results of the paper are very good and generalise past works while introducing interesting techniques like the derandomisation in order
to avoid the variable-time problem in quantum algorithms. The only issue of the paper is brushing some of the technicalities from quantum
algorithms aside, e.g., the quantum oracles employed are not defined and thus the overhead cost of nesting multiple quantum Monte Carlo
subroutines is ignored. I assume the authors took such costs to be constant, in which case the issue can probably be ignored. Still, it
would be interesting to state and compare the classical and quantum complexities in terms of other variables apart from $\varepsilon$. I wonder if a quadratic overhead on the oracle cost would show up, like in the other quantum algorithm for optimal stopping time.

---

> ### Author Rebuttal · Authors · 2026-03-31
>
> We thank the reviewer for the valuable suggestions. We also appreciate the recognition of the results and strong theoretical contributions, especially regarding the derandomization. Below, we summarize and address the questions raised.
>
> **Minor corrections:**
> We highlight and explain a few of the minor issues, and thank the reviewer for the other corrections that we implemented.
>
> * (6) $M_d$ is the number of Monte-Carlo repetitions to guarantee the $\varepsilon$ accuracy at time $d$. Compared to Algorithm 1, Algorithm 2 folds the $\varepsilon$ and corresponding pre-computed $M_d$ into the algorithm as parameters.
>
> * (8) $\mu_d(y_{<d})$ is defined as the expected value of the algorithm output $R_d(y_{<d}, \epsilon)$ which we show is independent of $\varepsilon$ as part of the theorem.
>
> * (16) When $n=0$, Delta is defined as a single $g_d$ term not a difference. This ensures for every $N$, the telescoping sums $\sum_{n=0}^N \Delta_d(y_{<d}, n)$ always gives a single term in $g_d$ at level $n=N$. This choice is the same as [1] and usual MLMC.
>
> **Quantum oracle model and complexity:**
> We thank the reviewer for this comment and agree that we need to add the quantum computation model and oracle access definition explicitly as a separate section of our paper. We assume the have the code model of [2] for the quantum-accelerated Monte Carlo implementation which is the same as the setup in [3] for the quantum algorithm in the case of a single nesting.
>
> Concretely, we assume the standard source-code access model: one has full algorithmic access to the classical sampling procedure.
> By that, we mean access to a Python script or pseudocode that generates samples from $p$, rather than access only through a black-box interface or API that returns samples. This is a standard assumption in the simulation literature, and it holds in all applications we are aware of.
>
> Given the above access, one can convert it into a reversible algorithm with only constant-factor overhead in time complexity [4], and then implement it as a quantum circuit
> $$
> U: |0\rangle \mapsto \sum_x \sqrt{p(x)} |x\rangle |\mathrm{garbage}_x\rangle,
> $$
> where the states $|\mathrm{garbage}_x\rangle$ may be arbitrary unit vectors. This state preparation model is also standard in the quantum computing literature, such as [2, 3].
>
> We require full access to the generator (rather than only to the samples it produces) because the quantum algorithm needs to modify the underlying random generation procedure so that it can be run on carefully chosen quantum superpositions. In [2], this is formalized as ``having the code'': access to a unitary quantum circuit with a fixed input such that, upon measuring its output and discarding some bits, we get a draw from the distribution, which is exact what the unitary $U$ gives in the conversion above. In our setting, the recursive routines and MLMC difference estimators built from them are exactly the randomized objects supplied to QAMC.
>
> We further assume that simulating one step of the trajectory and one function evaluation each have unit cost in our model. As explained in [2], in this model the complexity measure is the coherent implementation cost $C(A)$ of one execution of a randomized subroutine $A$
> if one coherent execution of a subroutine $A$ costs $C(A)$, then a call to QAMC still has per-query cost $\Theta(C(A))$, not $C(A)^2$. Therefore, the cost of nesting is already absorbed recursively into our definition of $C_d(\varepsilon)$ rather than an extra quadratic overhead. Moreover, [2] shows that the gate complexity of QAMC scales linearly with the gate complexity of implementing $A$.
>
> Ref:
> [1] Optimal randomized multilevel Monte Carlo for repeatedly nested expectations, Syed and Wang, ICML'23
>
> [2] Mean estimation when you have the source code; or, quantum Monte Carlo methods, Kothari and O'Donnell, SODA'23
>
> [3] Non-linear Quantum Monte Carlo, Blanchet, Hamoudi, Szegedy, and Wang, NeurIPS'25
>
> [4] Logical reversibility of computation, Bennett, IBM journal of Research and Development, 1973

---

> > ### Author Rebuttal · Reviewer_mPiR · 2026-04-05
> >
> > Thanks for your answers.

---

### Decision · Program_Chairs · 2026-04-30

**Decision:**

Accept (regular)

**Comment:**

The paper gives a novel quantum algorithm for repeatedly nested expectations (RNEs) that achieves nearly optimal error dependence $\tilde O(\varepsilon^{-1})$ (though with exponential dependence on number of nestings). Reviewers found the result significant and technically novel.
As requested in the reviews, I expect the authors to clarify quantum computation/oracle access model and carefully revise for typos and clarity.